# Deficiency of intestinal *Bmal1* prevents obesity induced by high-fat feeding

Fangjun Yu[1,2,5], Zhigang Wang[3,5], Tianpeng Zhang[1], Xun Chen[2], Haiman Xu[2], Fei Wang[2], Lianxia Guo[2], Min Chen[2], Kaisheng Liu[4✉] & Baojian Wu [1✉]

The role of intestine clock in energy homeostasis remains elusive. Here we show that mice with *Bmal1* specifically deleted in the intestine (*Bmal1*[iKO] mice) have a normal phenotype on a chow diet. However, on a high-fat diet (HFD), *Bmal1*[iKO] mice are protected against development of obesity and related abnormalities such as hyperlipidemia and fatty livers. These metabolic phenotypes are attributed to impaired lipid resynthesis in the intestine and reduced fat secretion. Consistently, wild-type mice fed a HFD during nighttime (with a lower BMAL1 expression) show alleviated obesity compared to mice fed *ad libitum*. Mechanistic studies uncover that BMAL1 transactivates the *Dgat2* gene (encoding the triacylglycerol synthesis enzyme DGAT2) via direct binding to an E-box in the promoter, thereby promoting dietary fat absorption. Supporting these findings, intestinal deficiency of *Rev-erbα*, a known BMAL1 repressor, enhances dietary fat absorption and exacerbates HFD-induced obesity and comorbidities. Moreover, small-molecule targeting of REV-ERBα/BMAL1 by SR9009 ameliorates HFD-induced obesity in mice. Altogether, intestine clock functions as an accelerator in dietary fat absorption and targeting intestinal BMAL1 may be a promising approach for management of metabolic diseases induced by excess fat intake.

[1] Institute of Molecular Rhythm and Metabolism, Guangzhou University of Chinese Medicine, Guangzhou, China. [2] College of Pharmacy, Jinan University, Guangzhou, China. [3] Department of Intensive Care Unit, First Affiliated Hospital of Jinan University, Guangzhou, China. [4] Shenzhen People's Hospital, The Second Clinical Medical College of Jinan University, Shenzhen, China. [5]These authors contributed equally: Fangjun Yu, Zhigang Wang. ✉email: kaisheng_liu@163.com; bj.wu@hotmail.com

Animals are rather efficient in absorbing dietary lipids (triacylglycerols or TAGs as the major form) and assimilating these nutrients as body fat. Intestinal fat absorption is a multistep process: dietary TAGs are first hydrolyzed in intestinal lumen to free fatty acids (FFAs) and monoacylglycerols (MAGs) which are subsequently taken up by the enterocytes, followed by resynthesis of TAGs in the cells and TAG packaging into chylomicrons for secretion to the circulation[1,2]. Resynthesis of TAGs can proceed via the MAG and glycerol-3-phosphate pathways, however, the MAG pathway plays a major role and provides about three-fourths of total TAGs synthesized in the intestine[3]. In the MAG pathway, MAGs are re-esterified with an FFA to form diacylglycerols (DAGs) by the enzymes monoacylglycerol acyltransferases (MOGATs), and DAGs are further esterified by diacylglycerol acyltransferases (DGATs) to produce TAGs. Notably, conversion of DAGs to TAGs is a rate-limiting step in TAG synthesis. It is therefore of no surprise that altered expression of intestinal DGATs is associated with deregulated dietary fat absorption and changes in susceptibility to diet-induced obesity and related comorbidities[4–6].

Circadian clocks orchestrate internal daily rhythms of molecular and cellular processes, leading to circadian variations in physiology and behaviors[7]. In mammals, the clock located in suprachiasmatic nucleus (SCN) of the brain acts as the central pacemaker coordinating the clocks present in other regions of the brain and in peripheral tissues[8]. These clocks are cell-autonomous oscillators built on several interconnected feedback loops[9]. In the core loop, the transcription factors BMAL1 and CLOCK associate to form a heterodimer that activates transcription of clock-controlled genes (CCGs) including PERs and CRYs[10]. As translated proteins of PERs and CRYs accumulate, they in turn inhibit the transcriptional activity of BMAL1/CLOCK complex and down-regulate the expression of CCGs[9]. Thereby, this delayed feedback generates ~24 h oscillations in expression of genes, including 43% of all protein-coding genes[11]. Additional feedback loops involve REV-REBs, RORs and D-box acting proteins (e.g., DBP and E4BP4) that serve to maintain robustness of the clock system[9]. REV-ERBs act as repressors, while RORs act as activators of BMAL1 expression, allowing a fine tuning of circadian rhythms[12].

Disruptions of circadian clock, by either genetic or environmental factors, profoundly affect energy metabolism and promote obesity[13–15]. For instance, Clock mutant mice are hyperphagic and obese with an attenuated feeding rhythm and show an increased susceptibility to atherosclerosis[13,16]. Social jet lag (a measure of circadian disruption) is associated with metabolic dysfunction and obesity[15]. However, the precise mechanisms underlying the metabolic abnormalities caused by clock malfunction are poorly understood. It remains to be answered to what extent the metabolic disorders result from abnormal behaviors or physiological outputs of the SCN or from loss of clock function in one or more peripheral tissues. Genetic studies also suggest a critical role for circadian clocks in regulation of dietary fat absorption, accounting for diurnal variations in lipid absorption processes[17–19]. However, the mice (with global deletion of a clock gene) in these studies had genetic deficiency of clock functions in all tissues including the brain, and consequently showed overall deregulation of feeding, sleep-wake cycle, hormonal release and locomotor activity. Thus the observed phenotypes (i.e., loss of rhythms in lipid absorption and lipid-related genes) may result from other factors such as arrhythmic feeding and gastric emptying besides the functional deficiency of intestine clock.

Excessive fat absorption (energy intake) contributes to obesity and associated metabolic disorders. Although circadian variations in fat absorption and other intestinal activities (e.g., DNA synthesis, cell proliferation, gastric and colonic motilities) have been noted, the precise role of the intestine clock in fat absorption and obesity development remains poorly defined and the underlying mechanisms are unaddressed[19,20]. It is challenging to separate the SCN effects from those of local clocks on intestinal functions using germline clock mutant animals[19]. To investigate the functional relevance of the intestine clock in dietary fat absorption and energy homeostasis, it is necessary to generate and test the mouse lines with intestine-specific deletion of the core clock gene.

Here, we show that mice with Bmal1 specifically deleted in the intestine (Bmal1[iKO] mice) have a normal phenotype on a chow diet. However, on a high-fat diet (HFD), Bmal1[iKO] mice are protected against development of obesity and related abnormalities such as hyperlipidemia and fatty livers. These metabolic phenotypes are attributed to impaired lipid resynthesis in the intestine and reduced fat secretion. Mechanistic studies uncover that BMAL1 transactivates the Dgat2 gene (encoding the triacylglycerol synthesis enzyme DGAT2), thereby promoting dietary fat absorption. Moreover, intestinal deficiency of Rev-erbα, a known BMAL1 repressor, enhances dietary fat absorption and exacerbates HFD-induced obesity and comorbidities. Therefore, we propose that intestine clock functions as an accelerator in dietary fat absorption and targeting intestinal BMAL1 may be a promising approach for management of metabolic diseases induced by excess fat intake.

## Results

**Deficiency of intestinal Bmal1 protects mice from high-fat diet-induced obesity.** To generate mice with intestine-specific loss of circadian clock function, we bred mice carrying a conditional Bmal1 allele (i.e., exon 8 floxed allele[21]) with mice expressing Cre recombinase under the control of the villin promoter (villin-Cre) that acts exclusively in intestinal epithelium[22]. To test the mice with Bmal1 specifically deleted in the intestine, we used male C57BL/6 mice homozygous for the Bmal1 conditional allele carrying single copy of the villin-Cre transgene (Bmal1[iKO] mice) and littermate mice homozygous for the Bmal1 conditional allele but not expressing Cre recombinase (control mice). Bmal1[iKO] mice were identified by PCR genotyping of genomic DNA from tail biopsies and had both floxed Bmal1 allele and Cre recombinase (Supplementary Fig. 1a). The mice showed the predicted loss of Bmal1 expression in the intestine but not in other tissues such as liver, kidney, brain and adipocytes (Supplementary Fig. 1b). Small intestine weight and length, intestinal motility, villus height, and crypt number of Bmal1[iKO] mice did not differ from control mice, suggesting normal development and differentiation of the intestinal epithelium (Supplementary Fig. 1c)[21]. As expected, targeted deletion of Bmal1 led to marked disruptions of rhythmic expression of other clock components (including Rev-erbα, Rev-erbβ, Per3, Npas2, Cry1, Cry2, Dbp and E4bp4) in the intestine (Supplementary Fig. 1d), indicating that the circadian clock function in the intestine is markedly compromised. It was not surprising that hepatic expression of these clock components did not change in Bmal1[iKO] mice (Supplementary Fig. 2).

The mice lacking intestinal Bmal1 gained weight normally on a chow (low-fat) diet (Fig. 1a). However, on a high-fat diet (HFD, 60% kcal from fat), Bmal1[iKO] mice gained less weight as compared to controls (Fig. 1a–c). After 10 weeks on HFD feeding, Bmal1[iKO] mice weighed 22% less than did control mice (Fig. 1a), and their fat mass was 46% lower with no significant difference in lean body weight (Fig. 1d). Supporting this, gonadal, mesenteric and inguinal white adipose tissues (gWAT, mWAT and iWAT) and brown adipose tissue (BAT) were all lighter in

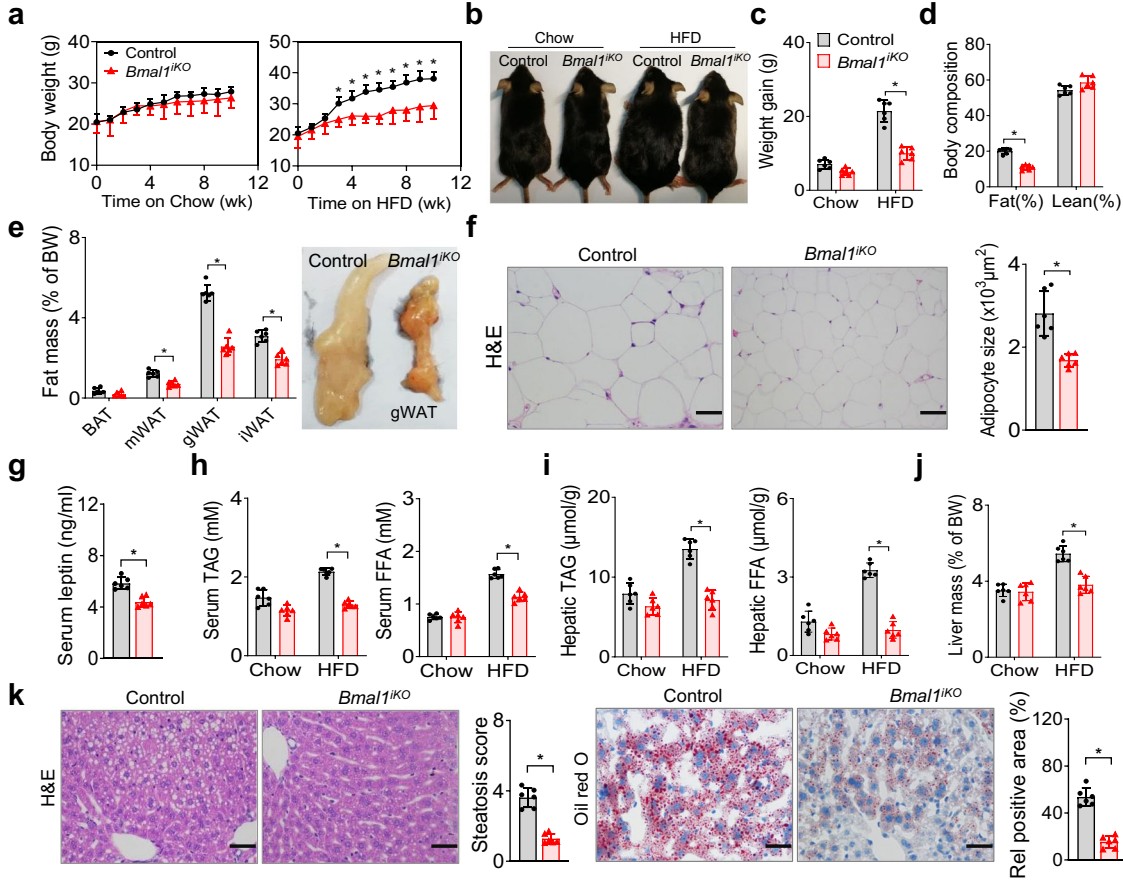

**Fig. 1 Deficiency of intestinal *Bmal1* protects mice from HFD-induced obesity. a** Body weight curves of *Bmal1^iKO* and control mice fed on a chow diet or HFD. **b** Representative images of *Bmal1^iKO* and control mice fed on chow diet or HFD for 10 weeks. **c** Body weight gain of *Bmal1^iKO* and control mice fed on chow diet or HFD for 10 weeks. Two-sided t test *p* values: 0.0501 (chow) and 0.0004 (HFD). **d** Body composition of *Bmal1^iKO* and control mice fed on HFD for 10 weeks. Two-sided t test *p* values: 0.0004 (Fat) and 0.0504 (Lean). **e** Fat percentages of *Bmal1^iKO* and control mice fed on HFD for 10 weeks. Images of gonadal white adipose tissues (gWAT) from *Bmal1^iKO* and control mice are shown in the right panel. **f** H&E staining of gWAT from *Bmal1^iKO* and control mice fed on HFD for 10 weeks. Adipocyte sizes are shown in the right panel. Two-sided t test *p* value: 0.0038. **g** Serum leptin levels in *Bmal1^iKO* and control mice fed on HFD for 10 weeks. Two-sided t test *p* value: 0.0055. **h** Serum TAG and FFA levels in *Bmal1^iKO* and control mice fed on chow diet or HFD for 10 weeks. Two-sided t test *p* values: 0.0538 (TAG, chow), <0.0001 (TAG, HFD); 0.8983 (FFA, chow), <0.0001 (FFA, HFD). **i** Hepatic TAG and FFA levels in *Bmal1^iKO* and control mice fed on chow diet or HFD for 10 weeks. **j** Liver masses of *Bmal1^iKO* and control mice fed on chow diet and HFD for 10 weeks. Two-sided t test *p* values: 0.7576 (chow) and 0.0016 (HFD). **k** H&E and oil red O staining of livers from *Bmal1^iKO* and control mice fed on HFD for 10 weeks. Two-sided t test *p* values: 0.0009 (left) and 0.0006 (right). Scale bar, 50 μm. All data points are mean ± SD (*n* = 6 biologically independent samples). For panels f (left) and k (micrographs), similar results were obtained in six independent experiments. In **a**, **e**, and **i**, statistical analysis was performed with two-sided t test or two-way ANOVA with Bonferroni post hoc test.*represents a *p* value of < 0.05.

*Bmal1^iKO* than in control mice (Fig. 1e). Histological analysis revealed adipocyte hypertrophy in control mice, and adipocytes from HFD-fed *Bmal1^iKO* mice were 41% smaller than those from controls (Fig. 1f). We also observed a lower concentration of serum leptin in *Bmal1^iKO* mice (Fig. 1g), which was expected considering a positive association of circulating leptin with adipose tissue mass[23]. Altogether, intestinal *Bmal1* deficiency protects mice from HFD-induced obesity.

We next examined whether *Bmal1^iKO* mice are protected from obesity-associated metabolic complications that can be induced by a HFD. *Bmal1^iKO* mice had lower levels of circulating TAG and FFA after HFD feeding for 10 weeks (Fig. 1h). They were also protected from hepatic steatosis as illustrated by decreased hepatic lipids, lower liver mass and histological examinations (Fig. 1i–k). Meanwhile, HFD-fed *Bmal1^iKO* mice had lower fasting blood glucose and insulin levels (Supplementary Fig. 3a). They showed better glucose tolerance and improved insulin sensitivity according to oral glucose tolerance test (OGTT) and insulin tolerance test (ITT) (Supplementary Fig. 3b). Therefore,

intestinal *Bmal1* deficiency also protects mice against hyperlipidemia, hepatic steatosis and glucose intolerance.

**Decreased dietary fat absorption in *Bmal1^iKO* mice.** To investigate the mechanisms underlying the phenotype of *Bmal1^iKO* mice, we first analyzed the energy balance. Food intake and feeding rhythm were similar between two genotypes (Supplementary Fig. 4a). *Bmal1^iKO* mice showed no changes in locomotor activity as compared to controls (Supplementary Fig. 4b). They did not show a difference either in energy expenditure normalized by body weight (Supplementary Fig. 4c). These data suggested that reduced adiposity and alleviated comorbidities in *Bmal1^iKO* mice were not due to an alteration in energy expenditure. We next examined intestinal absorption of dietary fat. On an HFD, *Bmal1^iKO* mice had a decreased efficiency in converting food into body mass, and the food efficiency was ~45% of controls (Fig. 2a). The mice also had lower levels of intestinal lipids (Fig. 2b), which was illustrated by oil red O staining (Fig. 2c). Conversely, fecal fat was higher in HFD-fed *Bmal1^iKO* mice than

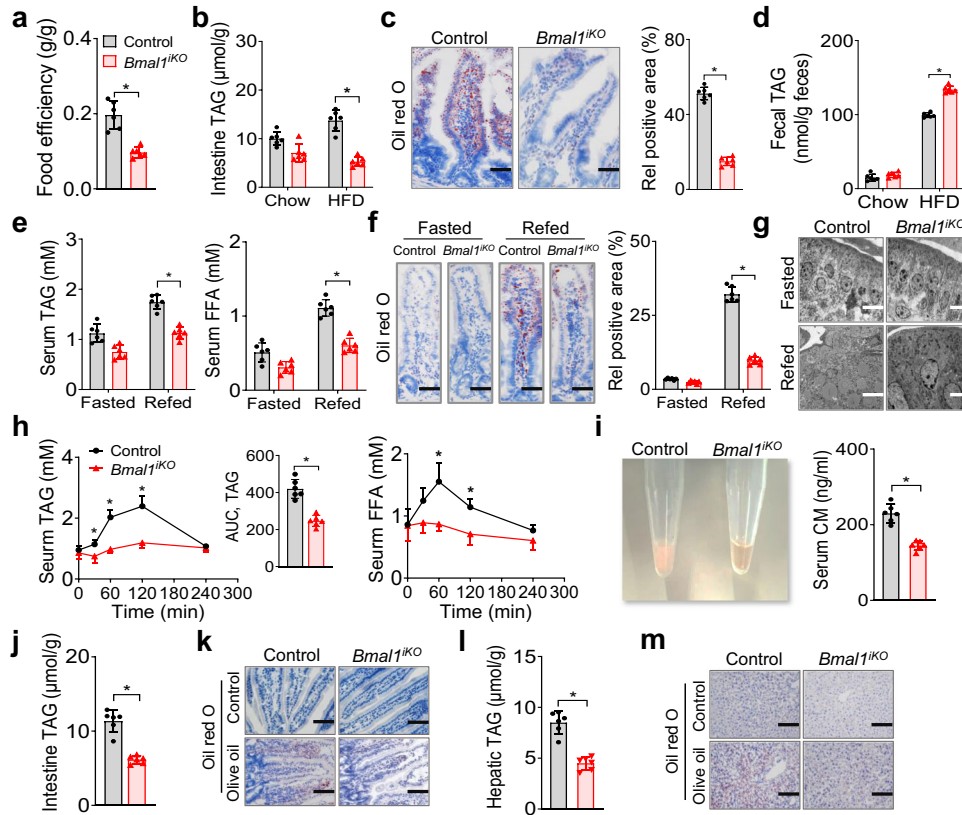

**Fig. 2 Decreased dietary fat absorption in Bmal1$^{iKO}$ mice. a** Food efficiency of Bmal1$^{iKO}$ and control mice during 10 weeks feeding on HFD. **b** Intestinal TAG levels in Bmal1$^{iKO}$ and control mice after 10 weeks feeding on chow or HFD. **c** Oil red O staining of proximal small intestine from Bmal1$^{iKO}$ and control mice after 10 weeks feeding on HFD. Scale bar, 50 μm. **d** Fecal TAG levels in Bmal1$^{iKO}$ and control mice after 10 weeks feeding on chow or HFD. **e** Serum TAG and FFA levels in fasted and refed Bmal1$^{iKO}$ and control mice. Mice were fasted for 12 h followed by HFD re-feeding for 1 h. **f** Oil red O staining of proximal small intestine from fasted and refed Bmal1$^{iKO}$ and control mice. Mice were treated as in **e**. Scale bar, 50 μm. **g** Electron microscopic images of proximal small intestine from fasted and refed Bmal1$^{iKO}$ and control mice. Scale bar, 5 μm. **h** Serum TAG (left) and FFA (right) levels in Bmal1$^{iKO}$ and control mice after oral gavage of olive oil (10 μl/g). The insert shows the AUC (area under the curve) values for serum TAG. **i** Image of serum samples (left) and measurements of serum CM (right). Samples were derived from Bmal1$^{iKO}$ and control mice at 2 h after oral gavage of olive oil (10 μl/g). **j** Intestinal TAG levels in Bmal1$^{iKO}$ and control mice at 2 h after oral gavage of olive oil. **k** Oil red O staining of proximal small intestine from Bmal1$^{iKO}$ and control mice at 2 h after oral gavage of olive oil. Scale bar, 100 μm. **l** Hepatic TAG levels in Bmal1$^{iKO}$ and control mice at 2 h after oral gavage of olive oil. **m** Oil red O staining of livers from Bmal1$^{iKO}$ and control mice at 2 h after oral gavage of olive oil. Scale bar, 100 μm. All data points are mean ± SD (n = 6 biologically independent samples). For **c** (left), **f** (left), **g**, **k** and **m**, similar results were obtained in six independent experiments. *represents a p value of < 0.05 (two-sided t test or two-way ANOVA with Bonferroni post hoc test). CM, chylomicrons.

in control mice (Fig. 2d). Furthermore, when fasted Bmal1$^{iKO}$ mice were refed with HFD for 1 h, they showed lower levels of serum TAG and FFA as well as decreased intestinal lipids compared to controls (Fig. 2e/f). This was consistent with a decreased number of and reduced size of lipid droplets in the enterocytes of Bmal1$^{iKO}$ mice (Fig. 2g). Additionally, when fasted mice were gavaged with olive oil, Bmal1$^{iKO}$ mice showed lower circulating TAG and FFA with reduced serum lactescence (decreased chylomicrons), and their TAG absorption was ~50% of controls (Fig. 2h/i). In line with this, intestinal and hepatic lipids were lower in olive oil-treated Bmal1$^{iKO}$ mice (Fig. 2j-m). These findings indicate that intestinal Bmal1 deficiency results in reduced absorption of dietary fat, potentially accounting for protection of the animals against HFD-induced obesity and comorbidities.

**Nighttime feeding ameliorates HFD-induced obesity in a BMAL1-dependent manner.** Because BMAL1 protein oscillates according to times of the day in various tissues including the intestine[21], we tested whether time-restricted feeding affects dietary fat absorption and diet-induced obesity. To this end, food

access of mice was restricted to the nighttime (from ZT13 to ZT22/23, activity phase) starting from the age of 6 weeks (Fig. 3a). The same regular amounts of daily calories were provided to the restricted feeding group and a control group with ad libitum access to food (Fig. 3b). On a chow diet, wild-type mice with restricted feeding did not differ in weight gain compared to mice fed ad libitum (Supplementary Fig. 5). However, mice fed a HFD during the dark period gained less weight and were lighter compared to mice fed ad libitum (Fig. 3c). The decreased body weight was associated with reduced masses of fat depots (gWAT and iWAT), smaller adipocytes and lower levels of circulating and hepatic lipids (Fig. 3c-g). These observations may be accounted for by reduced fat absorption due to lower expression of BMAL1 in the nighttime[21]. Intriguingly, the protective effects of nighttime feeding on HFD-induced weight gain and hyperlipidemia were lost in Bmal1$^{iKO}$ mice (Fig. 3c-g). We next examined dietary fat absorption in fasted mice by gavaging olive oil at each of different times of the day (ZT4, ZT8, ZT16 and ZT20), and 2 h post-gavage the levels of lipids in the plasma and tissues were measured. Oil gavage during the light phase generated higher levels of circulating, hepatic and intestinal lipids compared to oil gavage in the

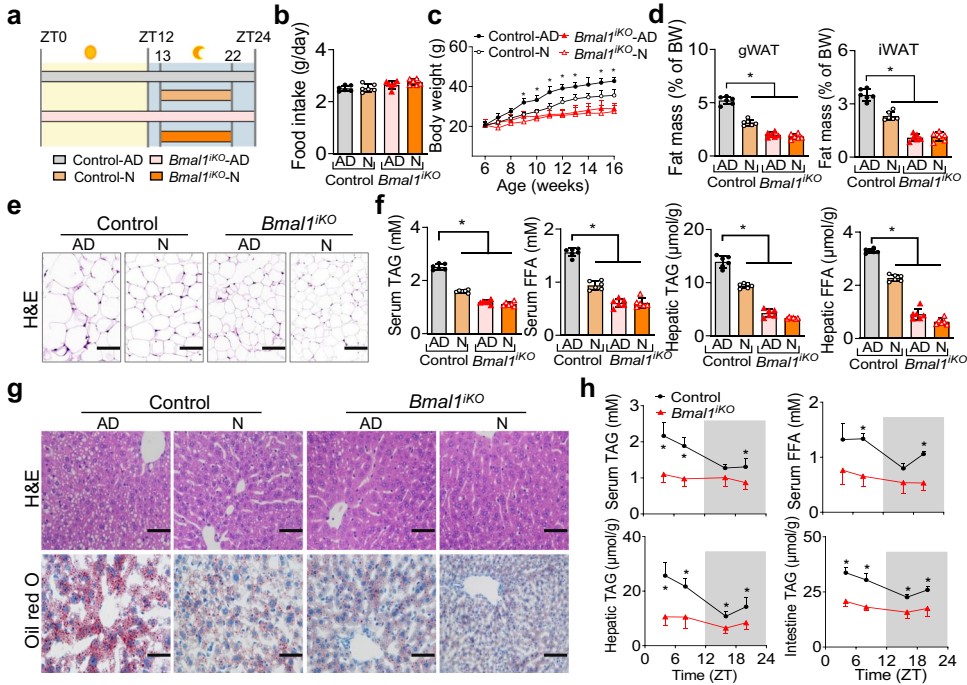

**Fig. 3 Nighttime feeding ameliorates HFD-induced obesity in a *Bmal1*-dependent manner. a** Schematic diagram of nighttime feeding and feeding *ad libitum* (on HFD). Mice with nighttime feeding had access to HFD for 9–10 h in the dark period from ZT13 to ZT22/23. AD, *ad libitum*, N, nighttime. **b** Daily food intake of *Bmal1iKO* and control mice fed *ad libitum* or with nighttime feeding. **c** Body weight curves of mice over 10 weeks feeding. *p* values (Control-AD *vs* Control-N, from left to right): 0.7144, 0.3415, 0.1321, 0.0121, 0.0104, 0.0153, 0.0023, 0.0055, 0.0512, 0.0031 and 0.0005 (two-way ANOVA and Bonferroni post hoc test). **d** Fat percentages of mice after 10 weeks feeding. *p* values (gWAT and iWAT, from left to right): < 0.0001, < 0.0001 and < 0.0001 (two-way ANOVA and Bonferroni post hoc test). **e** H&E staining of gWAT from mice after 10 weeks feeding. Scale bar, 50 μm. **f** Serum and hepatic TAG and FFA levels in mice after 10 weeks feeding. *p* values (serum TAG/FFA and hepatic TAG/FFA, from left to right): < 0.0001, < 0.0001 and < 0.0001 (two-way ANOVA and Bonferroni post hoc test). **g** H&E and oil red O staining of livers from mice after 10 weeks feeding. Scale bar, 50 μm. **h** Measurements of serum, hepatic and intestinal lipids in *Bmal1iKO* and control mice at 2 h after gavage of olive oil (10 μl/g) at ZT4, ZT8, ZT16 or ZT20. *p* values (serum TAG, from left to right): 0.0002, < 0.0001, 0.0547 and 0.0051 (two-way ANOVA and Bonferroni post hoc test). *p* values (serum FFA, from left to right): 0.0054, < 0.0001, 0.0601 and < 0.0001 (two-way ANOVA and Bonferroni post hoc test). *p* values (hepatic TAG, from left to right): < 0.0001, 0.0003, 0.0031, and 0.0066 (two-way ANOVA and Bonferroni post hoc test). *p* values (intestine TAG, from left to right): < 0.0001, < 0.0001, 0.0003 and 0.0005 (two-way ANOVA and Bonferroni post hoc test). All data points are mean ± SD (*n* = 6 biologically independent samples). For **e**, **g**, similar results were obtained in three independent experiments. *represents a *p* value of < 0.05.

dark phase, confirming circadian time-dependent fat absorption in the wild-type mice (Fig. 3h). Likewise, the time dependency of fat absorption after oil gavage ceased to exist in *Bmal1iKO* mice (Fig. 3h). Altogether, these findings support a role of intestinal BMAL1 in controlling dietary fat absorption in response to a HFD.

**Impaired intestinal lipid secretion in *Bmal1iKO* mice**. We wondered whether attenuated hyperlipidemia in *Bmal1iKO* mice is attributed to an increase in lipid uptake by peripheral tissues or a decrease in lipid secretion from the intestine. We thus examined fatty acid uptake by peripheral tissues by injecting [$^{14}$C]-oleic acid to mice. There were no differences in [$^{14}$C] levels in the serum, liver, heart, muscles or adipose tissues between *Bmal1iKO* and control mice (Fig. 4a, b). Tyloxapol, a lipoprotein lipase inhibitor, did not affect the reduction of serum TAG in HFD-fed *Bmal1iKO* mice (Fig. 4c). These data indicated that reduced serum lipids in *Bmal1iKO* mice was not due to an increase in uptake of circulating lipids by peripheral tissues. We then examined intestinal lipid secretion in *Bmal1iKO* and control mice by gavaging [$^{3}$H]-triolein. The serum [$^{3}$H] level was lower in *Bmal1iKO* than in control mice, suggesting reduced intestinal lipid secretion and absorption in *Bmal1iKO* mice (Fig. 4d). Consistently, decreased amounts of [$^{3}$H] were found in the small intestine, liver, heart, muscles and adipose tissues (BAT, gWAT, and iWAT) in *Bmal1iKO* mice, and

increased [$^{3}$H] in feces (Fig. 4e). However, there were no significant changes in [$^{3}$H] counts in the stomach and colon (Fig. 4e). Furthermore, *Bmal1iKO* mice showed lower levels of [$^{3}$H] in the proximal small intestine (duodenum and jejunum) compared to controls based on intestinal segment analysis (Fig. 4f). In addition, we analyzed the expression of lipogenesis-related and fatty acid mobilization-related genes in the liver and adipose tissue in two genotypes, and found that the expression of these genes was unaffected in *Bmal1iKO* mice (Supplementary Fig. 6). Taken together, these findings demonstrate impaired intestinal TAG secretion in *Bmal1iKO* mice, which underlies protection of the animals against HFD-induced hyperlipidemia and obesity.

**Intestinal ablation of *Bmal1* down-regulates DGAT2 expression**. Expression analysis of the main components responsible for dietary fat absorption revealed down-regulation of *Dgat2* gene (coding the TAG synthesis enzyme DGAT2) in the proximal intestine of *Bmal1iKO* mice (Fig. 5a and Supplementary Figs. 7 and 8). Consistently, DGAT2 protein was reduced in *Bmal1iKO* mice (Fig. 5b), while other TAG synthesis enzymes and the proteins for FFA uptake, lipid droplet formation and lipid oxidation, as well as chylomicron packaging (e.g., GPAT3, MOGAT2, DGAT1, FASN, CD36, PLIN3, PPARα, MTP and APOB48), were unaffected (Fig. 5b). Immunofluorescence assays

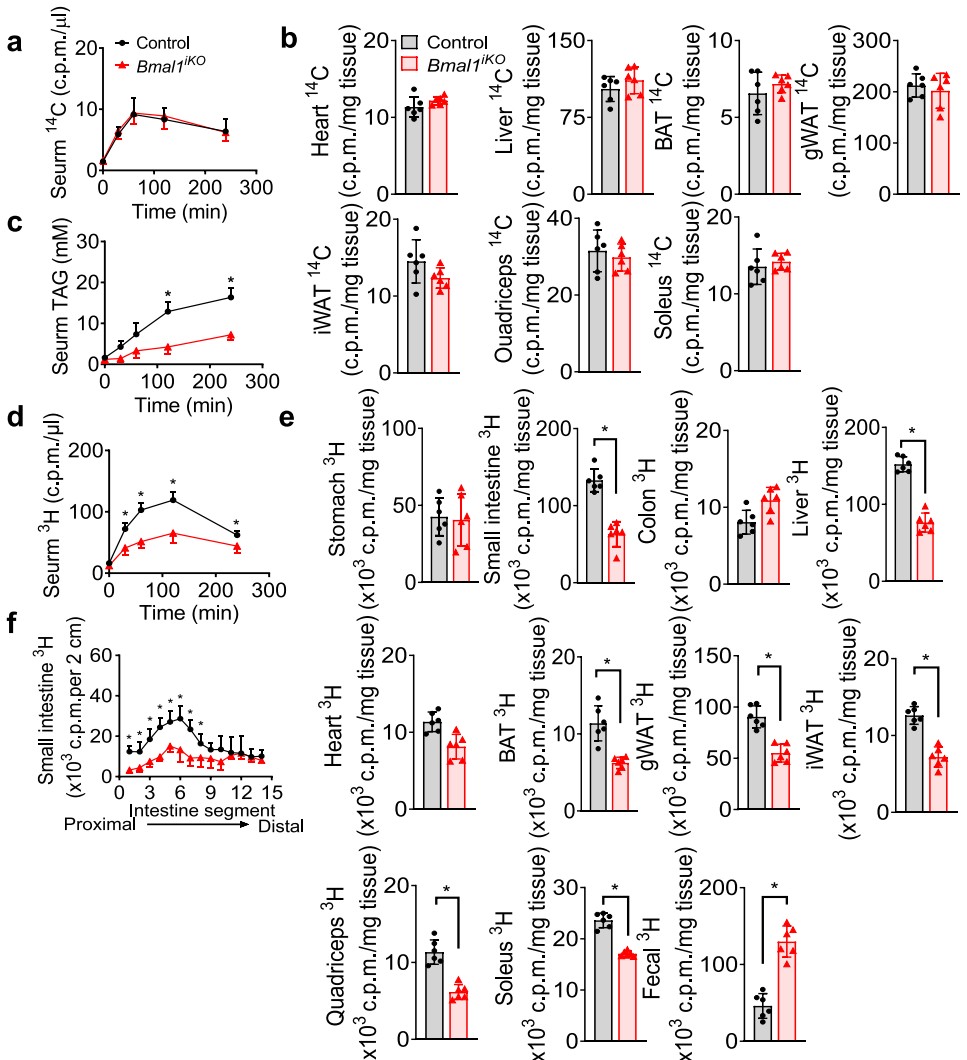

**Fig. 4 Impaired intestinal lipid secretion in *Bmal1^iKO* mice. a** Serum $^{14}$C radioactivities in *Bmal1^iKO* and control mice after i.p. injection of [$^{14}$C]-oleic acid. **b** $^{14}$C radioactivities in the tissues of *Bmal1^iKO* and control mice at 2 h after i.p. injection of [$^{14}$C]-oleic acid. **c** Serum TAG levels in tyloxapol (500 mg/kg, i.p., 30 min)-pretreated *Bmal1^iKO* and control mice after olive oil gavage (10 µl/g). *p* values (from left to right): 0.3179, 0.0019, 0.0106, < 0.0001 and <0.0001 (two-way ANOVA and Bonferroni post hoc test). **d** Serum $^{3}$H-radioactivities in *Bmal1^iKO* and control mice after gavage of [$^{3}$H]-triolein. *p* values (from left to right): 0.0716, 0.0004, <0.0001, <0.0001 and 0.0059 (two-way ANOVA and Bonferroni post hoc test). **e** $^{3}$H-radioactivities in the tissues and feces of *Bmal1^iKO* and control mice at 2 h after gavage of [$^{3}$H]-triolein. Two-sided t test *p* values: 0.8101 (stomach), <0.0001 (small intestine), 0.0594 (colon), <0.0001 (liver), 0.0511 (heart), 0.0004 (BAT), <0.0001 (gWAT), <0.0001 (iWAT), <0.0001 (quadriceps), <0.0001 (soleus) and <0.0001 (feces). **f** Distribution of dietary TAG in the small intestine of *Bmal1^iKO* and control mice at 2 h after gavage of [$^{3}$H]-triolein. *p* values (from left to right): <0.0001, 0.0040, 0.0009, <0.0001, 0.0008, 0.0015, 0.0002, 0.0246, 0.0895, 0.0842, 0.6833, 0.9999, 0.9278 and 0.1319 (two-way ANOVA and Bonferroni post hoc test). All data points are mean ± SD (*n* = 6 biologically independent samples). *represents a *p* value of < 0.05.

confirmed a reduction of intestinal DGAT2 protein in *Bmal1^iKO* mice as compared to control mice (Fig. 5c). Similar results were observed in HFD-fed mice (Supplementary Fig. 8). Furthermore, the mRNA and protein of DGAT2 varied according to times of the day in the intestine of control mice (Fig. 5d). *Bmal1* ablation markedly blunted the diurnal rhythm in intestinal DGAT2 in addition to reducing its expression level (Fig. 5d). These findings clearly demonstrate that ablation of intestinal *Bmal1* down-regulates the DGAT2 enzyme, which is tightly associated with dietary fat absorption.

We next examined the regulatory effects of BMAL1 on DGAT2 in mouse CT26, mouse primary intestinal epithelial cells, and human Caco-2 cells. Overexpression of *Bmal1* led to an increase in DGAT2 level in CT26 and primary intestinal epithelial cells,

whereas knockdown of *Bmal1* (by a specific siRNA) resulted in reduced expression of DGAT2 (Fig. 5e and Supplementary Figs. 9 and 10). By contrast, manipulation of *Bmal1* in CT26 cells did not affect other lipid absorption-related genes (Supplementary Fig. 9a). Similar observations were noted in Caco-2 cells (Fig. 5f and Supplementary Fig. 9b). As expected, *Dgat2* mRNA varied with the time in synchronized (serum-shocked) CT26 cells (Fig. 5g). *Bmal1* knockdown dampened the time-dependency of *Dgat2* in addition to reducing its level in the cells (Fig. 5g). Likewise, *DGAT2* mRNA oscillated in a time-dependent manner in synchronized Caco-2 cells, and its oscillation was attenuated by *BMAL1* silencing (Fig. 5h). These cell-based results support a positive role of BMAL1 in circadian regulation of intestinal DGAT2.

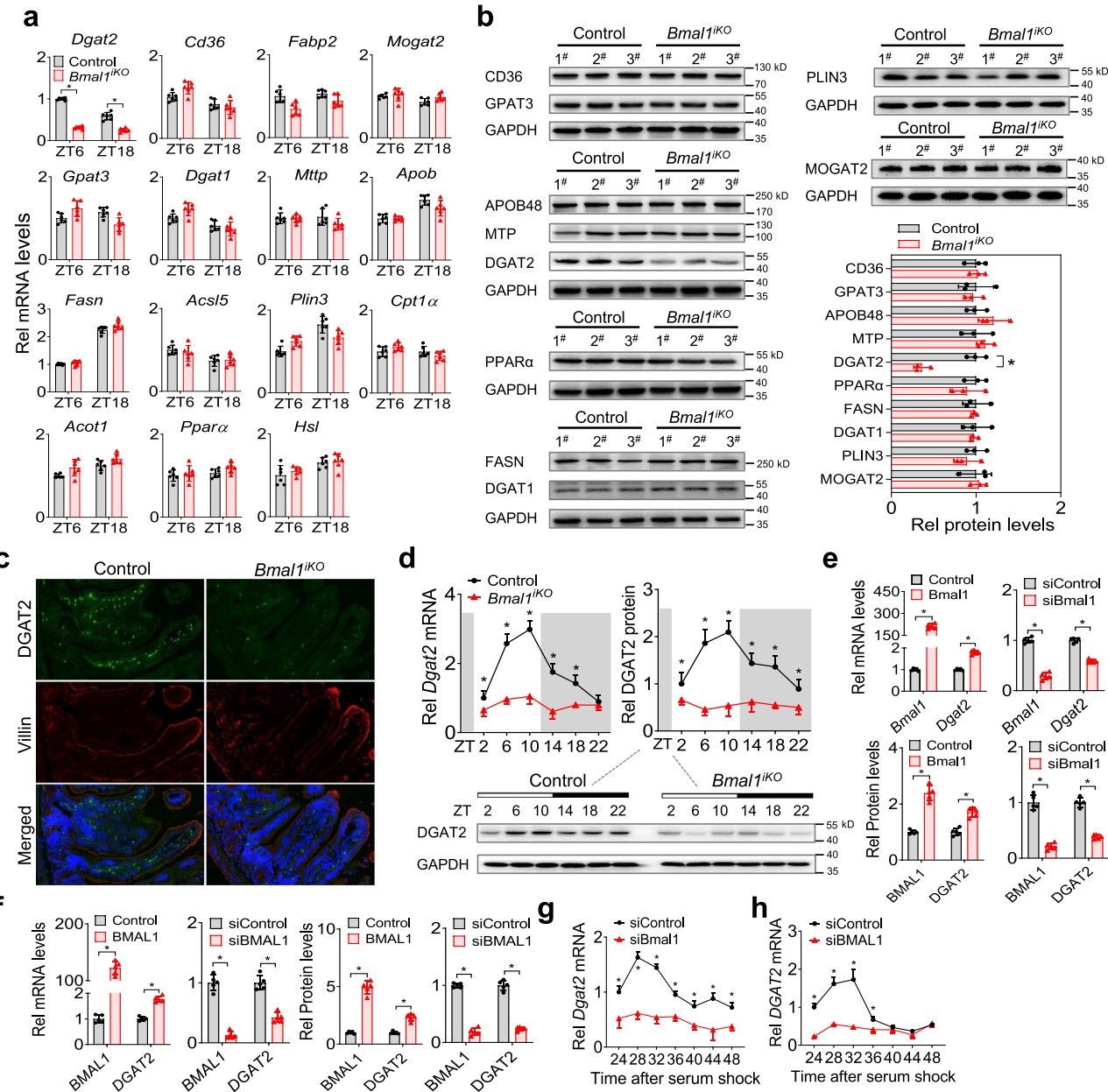

**Fig. 5 Intestinal ablation of *Bmal1* down-regulates DGAT2 expression. a** qPCR analyses of the main components responsible for dietary fat absorption in small intestines of *Bmal1^iKO^* and control mice at ZT6 and ZT18. Data are mean ± SD (*n* = 6 biologically independent samples). Two-sided t test *p* values (*Dgat2*): < 0.0001 and < 0.0001. **b** Western blotting analyses of the main components responsible for dietary fat absorption in small intestines of *Bmal1^iKO^* and control mice at ZT6. Western blot strips (one or more target proteins and a loading control) were cut from one gel. In particular, the blot of MOGAT2 was stripped and re-probed for GAPDH as a loading control. Data are mean ± SD (*n* = 3 biologically independent samples). Two-sided t test p value (DGAT2): <0.0001. **c** Double immunofluorescence staining of DGAT2 (green) and villin (red) in small intestines of *Bmal1^iKO^* and control mice at ZT6. **d** Diurnal DGAT2 mRNA and protein expression in small intestines of *Bmal1^iKO^* and control mice. Western blot strips (a target protein and a loading control) were cut from one gel. Data are mean ± SD (*n* = 6 biologically independent samples). **e** *Dgat2* mRNA and DGAT2 protein levels in CT26 cells transfected with Bmal1 plasmid or siBmal1 or control. Data are mean ± SD (*n* = 5 biologically independent samples). **f** *DGAT2* mRNA and DGAT2 protein levels in Caco-2 cells transfected with BMAL1 plasmid or siBMAL1 or control. Data are mean ± SD (*n* = 5 biologically independent samples). **g** *Bmal1* knockdown attenuates *Dgat2* rhythmicity in synchronized CT26 cells. Data are mean ± SD (*n* = 5 biologically independent samples). **h** *BMAL1* knockdown attenuates *DGAT2* rhythmicity in synchronized Caco-2 cells. Data are mean ± SD (*n* = 5 biologically independent samples). For **b**, **c** and **d** (bottom), similar results were obtained in three independent experiments. In panels d–h, statistical analysis was performed with two-sided t test or two-way ANOVA with Bonferroni post hoc test. *represents a *p* value of < 0.05.

**BMAL1 regulates the transcription of *Dgat2*.** Parallel changes in mRNA and protein of DGAT2 prompted us to investigate whether BMAL1 (also known as a transcription factor) regulates *Dgat2* using a transactivation mechanism. In luciferase reporter assays, BMAL1 dose-dependently induced the promoter activity

of *Dgat2* (Fig. 6a). Sequence analysis predicted three potential E-boxes (i.e., the putative DNA motifs for BMAL1 binding and action) in the promoter region of *Dgat2* gene (Fig. 6b). Truncation and mutation experiments identified an E-box (i.e., −340/−328 bp) in the *Dgat2* promoter, which was actually responsible

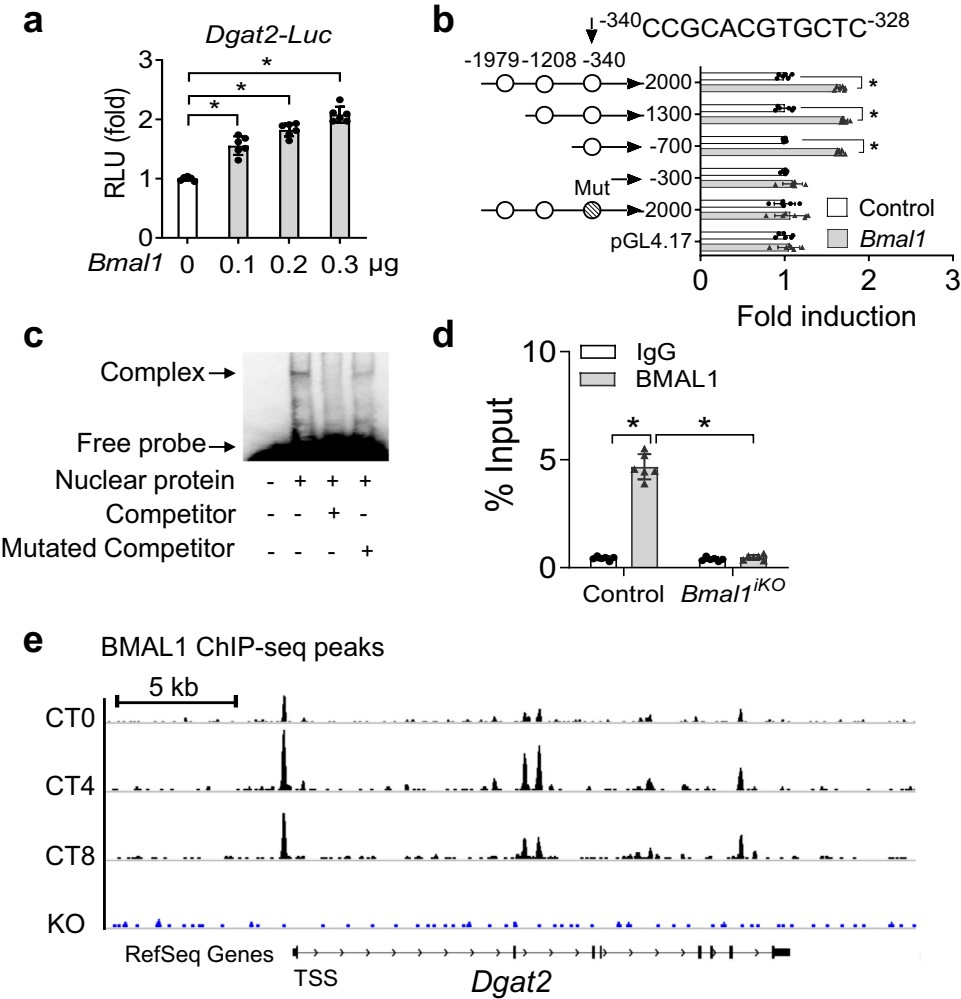

**Fig. 6 BMAL1 regulates the transcription of *Dgat2*. a** *Bmal1* dose-dependently induced *Dgat2* transcription in luciferase reporter assays. NIH3T3 cells were transfected with *Dgat2*-luc reporter and different amounts of *Bmal1* plasmid. After 24 h transfection, luciferase reporter activities were measured. *p* values (from left to right): <0.0001, <0.0001, <0.0001, and <0.0001 (one-way ANOVA and Bonferroni post hoc test). **b** Effects of *Bmal1* on the activities of various versions of *Dgat2*-luc reporters. Two-sided *t* test *p* values (from top to bottom): <0.0001, <0.0001, <0.0001, 0.1811, 0.4680, and 0.4431. **c** EMSA assays showing a direct interaction between BMAL1 protein and *Dgat2* E-box element. **d** ChIP assays showing significant recruitment of BMAL1 protein to *Dgat2* E-box in the small intestine of control mice, but no such recruitment in the small intestine of *Bmal1*iKO mice. *p* values (from left to right): <0.0001 and 0.2031 (two-way ANOVA and Bonferroni post hoc test). **e** ChIP-seq showing enrichment of BMAL1 protein to *Dgat2* promoter (data obtained from GEO database with an accession number of GSE130505). All data points are mean ± SD (*n* = 6 biologically independent samples). *represents a *p* value of <0.05.

for BMAl1 action (Fig. 6b). Electrophoretic mobility shift assay (EMSA) confirmed a direct interaction of BMAL1 protein with the identified E-box of *Dgat2* (Fig. 6c). In addition, chromatin immunoprecipitation (ChIP) assays revealed significant recruitment of intestinal BMAL1 protein to the E-box of *Dgat2* (Fig. 6d). However, such recruitment was lost in the intestine from *Bmal1*iKO mice (Fig. 6d). Likewise, ChIP-seq indicated circadian time-dependent enrichment of BMAL1 protein to *Dgat2* promoter in wild-type mice and loss of such enrichment in *Bmal1*-null mice (Fig. 6e, GSE130505). Altogether, BMAL1 drives the transcription of *Dgat2* through direct binding to an E-box element in the gene promoter.

**Intestinal deficiency of the BMAL1 repressor *Rev-erbα* promotes HFD-induced obesity.** *Rev-erbα* is a known negative regulator of BMAL1 and also an integral component of circadian clock[24]. We thus wondered whether intestinal REV-ERBα plays an opposite role in regulating HFD-induced obesity. To generate mice with intestine-specific deletion of *Rev-erbα* (*Rev-erbα*iKO

mice), we bred mice carrying a conditional *Rev-erbα* allele (i.e., exons 2-6 floxed allele) with mice expressing *villin-Cre* (Supplementary Fig. 11a). *Rev-erbα*iKO mice showed a predicted increase in intestinal BMAL1 expression (Supplementary Fig. 11b). Food intake and feeding rhythm as well as locomotor activities were not different in the mice (Supplementary Fig. 11c, d). Interestingly, *Rev-erbα*iKO mice gained more weight on a HFD as compared to controls (Fig. 7a, b). After 10 weeks on HFD feeding, *Rev-erbα*iKO mice were 18% heavier than did control mice (Fig. 7a, b), and their fat mass was 37% higher with no significant difference in lean body weight (Fig. 7c). Increased adiposity in *Rev-erbα*iKO mice was accompanied by hyperlipidemia and more severe hepatic steatosis (Fig. 7d–f). HFD-fed *Rev-erbα*iKO mice also had higher TAG level in the small intestine (Fig. 7g). In addition, circulating, hepatic and intestinal lipids were higher in fasted *Rev-erbα*iKO mice than in controls after gavaging olive oil (Supplementary Fig. 11e–g). Thus, intestinal *Rev-erbα* deficiency enhances dietary fat absorption and exacerbates HFD-induced obesity and associated metabolic disorders.

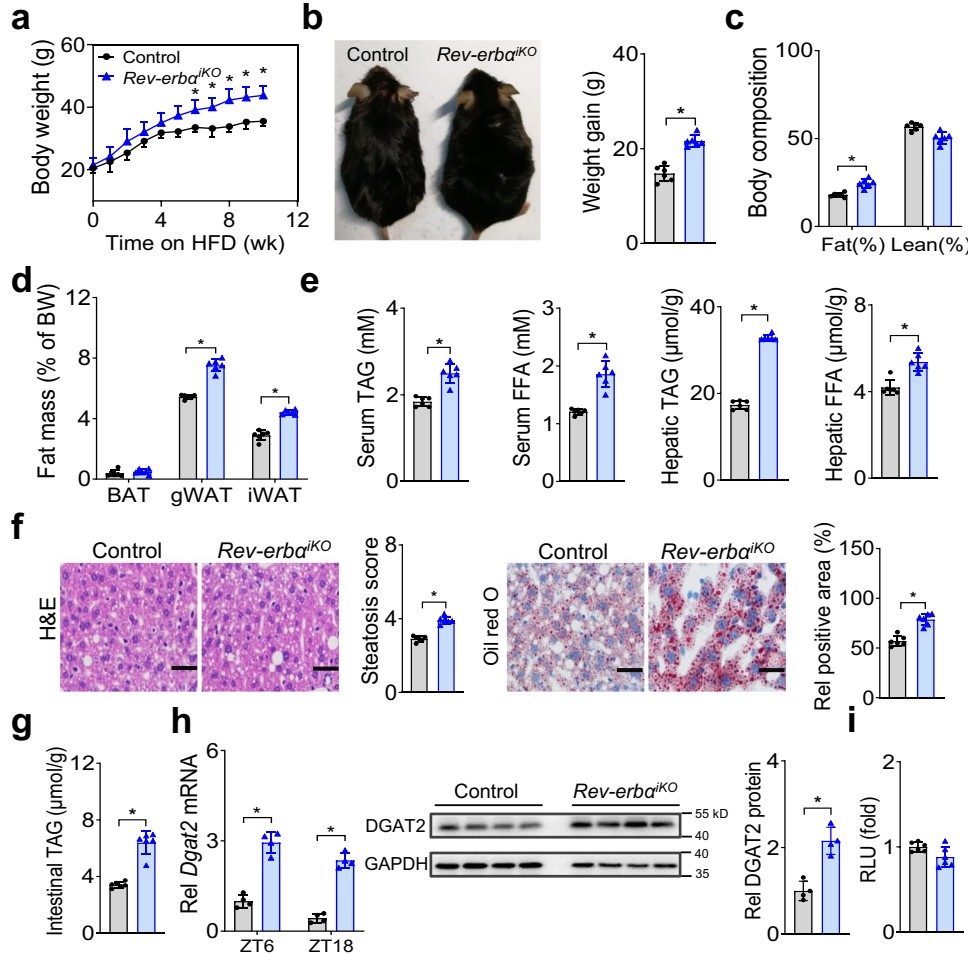

**Fig. 7 Intestinal deficiency of the BMAL1 repressor *Rev-erbα* promotes HFD-induced obesity. a** Body weight curves of *Rev-erbα^iKO* and control mice fed on HFD. *p* values (from left to right): 0.2430, 0.3615, 0.2505, 0.1701, 0.1694, 0.0513, 0.0423, 0.0066, 0.0041, 0.0056 and 0.0035 (two-way ANOVA and Bonferroni post hoc test). **b** Representative image (left) and body weight gain (right) of *Rev-erbα^iKO* and control mice after 10 weeks feeding on HFD. Two-sided t test *p* value: < 0.0001. **c** Body composition of *Rev-erbα^iKO* and control mice fed on HFD for 10 weeks. Two-sided t test *p* values: 0.0005 (Fat) and 0.0568 (Lean). **d** Fat percentages of *Rev-erbα^iKO* and control mice fed on HFD for 10 weeks. Two-sided t test *p* values: 0.2907 (BAT), <0.0001 (gWAT) and < 0.0001 (iWAT). **e** Serum and hepatic TAG and FFA levels in *Rev-erbα^iKO* and control mice fed on HFD for 10 weeks. Two-sided t test *p* values (from left to right): < 0.0001, < 0.0001, < 0.0001 and 0.0003. **f** H&E and oil red O staining of livers from *Rev-erbα^iKO* and control mice fed on HFD for 10 weeks. Two-sided t test *p* values (from left to right): < 0.0001 and < 0.0001. Scale bar, 50 μm. **g** Intestinal TAG levels in *Rev-erbα^iKO* and control mice fed on HFD for 10 weeks. Two-sided t test *p* value: < 0.0001. **h** mRNA and protein expression of DGAT2 in small intestines of *Rev-erbα^iKO* and control mice. Western blot strips (a target protein and a loading control) were cut from one gel. Data are mean ± SD (*n* = 4 biologically independent samples). Two-sided t test *p* values (from left to right): < 0.0001, < 0.0001 and 0.0010. **i** *Rev-erbα* (blue bar) has no effect on *Dgat2* transcription in luciferase reporter assay. Two-sided t test *p* value: 0.2604. In all panels except h, data are mean ± SD (*n* = 6 biologically independent samples). For panel f (micrographs), similar results were obtained in six independent experiments. For panel h (gel), similar results were obtained in four independent experiments. *represents a *p* value of < 0.05.

Expression analysis of the main components responsible for dietary fat absorption revealed up-regulation of *Dgat2* gene in *Rev-erbα^iKO* mice, while others were unaffected (Fig. 7h and Supplementary Fig. 11h). In line with the mRNA change, the protein of DGAT2 was elevated in the mice (Fig. 7h). Because REV-ERBα functions as a transcriptional repressor, we tested whether it can directly regulate *Dgat2*. In luciferase reporter assays, REV-ERBα had no effects on the promoter activity of *Dgat2* (Fig. 7i). No RevRE element (i.e., the putative DNA motif for REV-ERBα binding and action) was found in the promoter of *Dgat2* according to sequence analysis. Thus, direct regulation of *Dgat2* by REV-ERBα was unlikely. Given that REV-ERBα is a repressor of BMAL1, which is an activator of DGAT2 (Fig. 6), it was reasoned that REV-ERBα negatively regulated *Dgat2* expression through repressing its activator BMAL1. This was

supported by the fact that the regulatory effects of REV-ERBα on the transcription and expression of *Dgat2* were attenuated when *Bmal1* was silenced (Supplementary Fig. 11i).

**Small-molecule targeting of REV-ERBα/BMAL1 ameliorates HFD-induced obesity.** Given the critical role of REV-ERBα/BMAl1 in the development of HFD-induced obesity, it is of interest to test whether REV-ERBα/BMAL1 can be targeted to prevent this metabolic disorder. To this end, we examined the effects of the small molecule SR9009 on obesity development in mice fed a HFD. SR9009 is a known REV-ERBα agonist that can decrease BMAL1 expression[25,26]. After oral gavage, SR9009 had very limited systemic exposure (plasma levels were below 0.3 μg/ml), however, drug concentrations in the intestine were

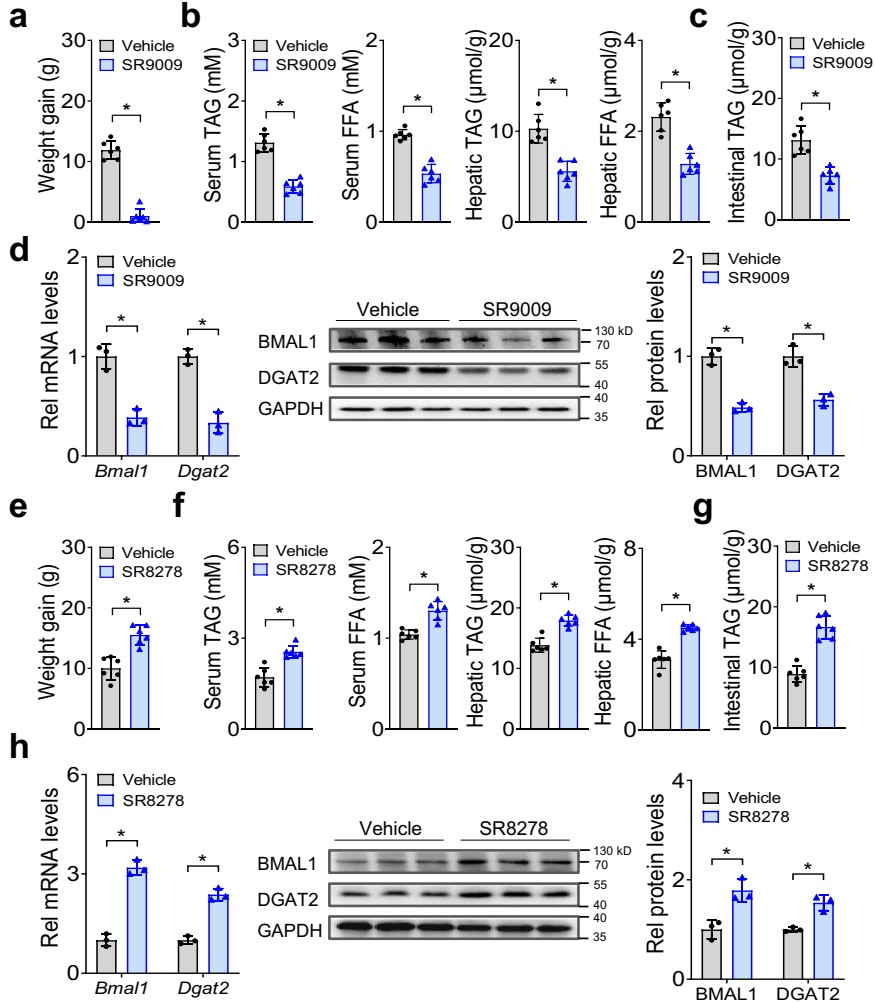

**Fig. 8 Small-molecule targeting of REV-ERBα/BMAL1 ameliorates HFD-induced obesity. a** Effects of oral SR9009 on body weight gain in HFD-fed mice (wild-type). 6-week-old mice were fed HFD for 4 weeks. The mice continued on HFD and were gavaged with SR9009 (100 mg/kg) or vehicle once daily for 4 weeks. Body weights were measured after 4-weeks drug treatment. Two-sided t test *p* value: < 0.0001. **b** Effects of oral SR9009 on serum and hepatic TAG and FFA levels. Two-sided t test *p* values (from left to right): <0.0001, <0.0001, <0.0001 and <0.0001. **c** Effects of oral SR9009 on intestinal TAG level. Two-sided t test *p* value: 0.0003. **d** Effects of oral SR9009 on BMAL1 and DGAT2 expression in small intestine of mice. Western blot strips (two target proteins and a loading control) were cut from one gel. Two-sided t test *p* values (from left to right): 0.0020, 0.0009, 0.0007 and 0.0033. **e** Effects of oral SR8278 on body weight gain in HFD-fed mice (wild-type). 6-week-old mice were fed a HFD for 4 weeks. The mice continued on HFD and were gavaged with SR8278 (100 mg/kg) once daily for 4 weeks. Body weights were measured after 4-weeks drug treatment. Two-sided t test *p* value: 0.0003. **f** Effects of oral SR8278 on serum and hepatic TAG and FFA levels. Two-sided t test *p* values (from left to right): 0.0003, 0.0002, <0.0001 and <0.0001. **g** Effects of oral SR8278 on intestine TAG levels. Two-sided t test *p* value: < 0.0001. **h** Effects of oral SR8278 on BMAL1 and DGAT2 expression in small intestine of mice. Western blot strips (two target proteins and a loading control) were cut from one gel. Two-sided t test *p* values (from left to right): 0.0002, 0.0004, 0.0097 and 0.0045. In all panels except **d** and **h**, data are mean ± SD (*n* = 6 biologically independent samples). In panels **d** and **h**, Data are mean ± SD (*n* = 3 biologically independent samples). For panels **d** (middle) and **h** (middle), similar results were obtained in three independent experiments. *represents a *p* value of < 0.05.

reasonably high (C<sub>max</sub>> 10 μg/ml) (Supplementary Fig. 12a). We initiated the study with 10-week-old wild-type mice that had been maintained on HFD for 4 weeks. The mice continued on HFD and we initiated oral gavage of SR9009 once daily for 4 weeks. There was no difference in food intake between SR9009- and vehicle-treated mice (Supplementary Fig. 12b). SR9009 treatment significantly decreased the body weight gain in mice (Fig. 8a). This was accompanied by alleviated hyperlipidemia and hepatic steatosis (Fig. 8b). SR9009-treated mice also had lower TAG level in the small intestine (Fig. 8c). Furthermore, SR9009 reduced the expression levels of intestinal DGAT2 in addition to BMAL1 (Fig. 8d), but had no effects on expression of other factors involved in dietary fat absorption (Supplementary Fig. 12c).

Because SR9009 (injected intraperitoneally) was previously shown to impact body lipid metabolism[26], we analyzed the expression of lipid metabolism-related genes in major metabolic organs (Supplementary Fig. 12d). SR9009 gavage did not alter the expression of lipogenic genes (*Srebp1c, Fasn* and *Scd1*) in the liver, TAG synthesis genes (*Mogat2, Dgat1, Dgat2, Gpat3, Plin3* and *Hsl*) in the WAT, and genes involved in fatty acid oxidation (*Cpt1β, Ucp1* and *Pgc1α*) in the skeletal muscle (Supplementary Fig. 12d). Thus, the effects of oral SR9009 may be limited to the gut. We also initiated a study with 6-week-old mice that had been kept on chow diet. The mice were then switched to HFD feeding and gavaged with SR9009 once daily for 8 weeks. Likewise, in this experimental scheme, SR9009-treated mice showed reduced

weight gain as well as less severe hyperlipidemia and hepatic steatosis as compared to vehicle treatment (Supplementary Fig. 13). Additionally, we observed the opposite effects of SR8278 (a known REV-ERBα antagonist[27]) on HFD-fed mice, including exacerbated obesity, aggravated hyperlipidemia and increased intestinal DGAT2 (Fig. 8e–h). Taken together, these results corroborate that REV-ERBα/BMAL1 plays an essential role in development of HFD-induced obesity and indicate that targeted inhibition of BMAL1 by REV-ERBα activation ameliorates HFD-induced obesity.

## Discussion

Circadian clock machinery is present in virtually all tissues. In addition to the SCN clock, clocks in peripheral tissues such as the liver, adipocytes, and skeletal muscle are involved in energy balance and lipid homeostasis[28–30]. Hepatocyte-specific deletion of *Bmal1* in mice promotes hyperlipidemia by modulating lipoprotein production and biliary cholesterol excretion[28]. Adipocyte-specific ablation of *Bmal1* results in obesity in mice due to an attenuated rhythm in food intake (increased food intake in daytime)[29]. By contrast, specific disruption of *Bmal1* in the skeletal muscle prevents lipid deposition and HFD-induced obesity by increasing oxidative capacity[30]. Muscle-specific loss of *Bmal1* is also associated with impaired muscle triglyceride biosynthesis and accumulation of bioactive lipids[31]. These distinct contributions of peripheral clocks to lipid metabolism and homeostasis highlight that global knockout mice (with clock deficiency in all tissues) are insufficient to delineate the clock function in a particular tissue due to interferences from other tissues.

By using intestine-specific knockout mice, we have been able to clarify the role of intestine clock in dietary fat absorption. We reveal that the intestine clock (BMAL1) functions to promote dietary fat absorption by regulating the TAG synthesis enzyme DGAT2. This confers an advantage in times of food deprivation, but contributes to obesity and associated metabolic disorders when dietary fat is abundant (e.g., HFD feeding). Thus, intestinal clock dysfunction (*Bmal1* deficiency) protects mice against HFD-induced obesity. It is noted that on a chow diet, mice lacking intestinal *Bmal1* have a normal phenotype (Fig. 1). This supports the notion that dietary fat absorption is remarkably efficient in animals (e.g., complete or near complete fat absorption when fed a normal diet) and highlights a crucial role of intestine clock (BMAL1) in nutrient absorption in the case of energy oversupply. Given a role of circadian oscillator in dietary fat absorption, it is of no surprise that HFD-induced obesity depends on the feeding time. Mice had less efficient intestinal fat absorption and decreased susceptibility to HFD-induced obesity when a same regular amount of food was restricted to the nighttime, consistent with lower expression of BMAL1 protein in the nighttime[21]. We attempted to test whether time-restricted feeding during the daytime affects HFD-induced obesity. Because mice with daytime food restriction do not consume a regular amount of food[32,33], we adopted an initial adjustment period (days 1–15) that allowed mice to adapt to the time-restricted access to food before introducing mice to a same amount of HFD[29]. Wild-type mice fed during the daytime gained more weight and were heavier compared to mice fed *ad libitum* and mice fed during the nighttime (Supplementary Fig. 14). However, these differences were not seen in *Bmal1^{iKO}* mice (Supplementary Fig. 14), supporting a critical role of intestinal BMAL1 in circadian fat absorption and HFD-induced obesity. Contrasting with a prior report that mice absorb lipids more efficiently in the nighttime (activity phase) based on short-term (≤1 h) in situ and in vitro experiments[17], our data suggest more efficient fat absorption

during the light period in HFD-fed mice and we did not observe a feeding time-dependency in fat absorption or body weight in mice fed a normal diet (Supplementary Fig. 5). We argue that the short-term experiments may not accurately measure the dietary fat absorption on a daily basis because the extent of fat absorption depends not only on the rate of absorption but also on the time of absorption (> 6 h per day)[34]. Importantly, our findings agree with the recent perspectives on humans that nighttime consumption of low energy food is harmless but high calorie intake in the evening contributes to weight gain and poor health[35].

We have shown that activation of intestinal REV-ERBα by SR9009 (a synthetic agonist) down-regulates BMAL1 expression in the intestine to reduce fat absorption and ameliorate HFD-induced obesity (Fig. 8), providing a unique strategy for obesity prevention and control. REV-ERBα has been also proposed as a therapeutic target for many other disorders including hyperglycemia, heart failure, cancers and inflammatory diseases[36]. Despite established pharmacological effects in animals, SR9009 and other synthetic agonists (e.g., SR9011 and GSK4112) are concerned with poor pharmacokinetic properties (e.g., low systemic exposure and short half-lives) and potential adverse effects on off-target tissues (e.g., the SCN)[26,36]. Thus their drugability is severely compromised as illustrated by the lack of clinical trials on these molecules[36]. Nevertheless, the intestine as a target tissue is advantageous in drug development considering that it is easier to achieve high local drug exposure in the intestine but low systemic exposure (so as to avoid the adverse effects on SCN and other vital tissues) via oral administration. For instance, ezetimide is a cholesterol-lowering drug that targets the sterol transporter Niemann-Pick C1-Like 1 (NPC1L1) in the intestine and has low systemic bioavailability[37,38]. It is noteworthy that the protective effects of SR9009 against HFD-induced obesity may depend not only on REV-ERBα/BMAL1 but also on REV-ERBβ/BMAL1 because SR9009 is a dual agonist of REV-ERBα and REV-ERBβ, both of which repress *Bmal1* transcription and expression[26,39], and REV-ERBβ is also expressed in the intestine[40].

We observed lower total cholesterol level in HFD-fed *Bmal1^{iKO}* than in control mice (Supplementary Fig. 15a). Thus, intestinal deficiency of *Bmal1* also protects mice against hypercholesterolemia induced by a HFD. We analyzed the expression of major factors involved in cholesterol metabolism, and found that intestinal *Abca1* was down-regulated and its rhythm was blunted in *Bmal1^{iKO}* mice (Supplementary Fig. 15b). *Abca1* encodes the ABCA1 protein that mediates the secretion of free cholesterol into apolipoprotein A-1 to form high-density lipoprotein, thereby playing a critical role in cholesterol homeostasis[41]. Luciferase reporter assays indicated that BMAL1 positively regulates *Abca1* transcription (Supplementary Fig. 15c). These findings seem to be consistent with a prior study in which CLOCK, the BMAL1 binding partner, regulates ABCA1 expression using an indirect mechanism involving the transcription factor USF2[16]. It is therefore reasonable to propose that BMAL1 promotes intestinal cholesterol secretion through regulating ABCA1 expression. We have also shown that intestinal *Bmal1* deficiency results in a lower level of fasting plasma glucose as well as improved glucose tolerance and insulin sensitivity in HFD-fed *Bmal1^{iKO}* mice compared to controls (Supplementary Fig. 3), however, the underlying mechanism was unexplored. A prior study reports that circadian clock may regulate carbohydrate transporters (e.g., SGLT1, GLUT2 and GLUT5) to affect intestinal glucose absorption[17]. We did observe expression changes (i.e., slight increases) in intestinal *Sglt1*, *Glut2*, and *Glut5* in *Bmal1^{iKO}* mice (Supplementary Fig. 15d). However, this apparently is not a cause of reduced fasting plasma glucose. We argue that the improved glucose profiles in HFD-fed *Bmal1^{iKO}* mice may result from amelioration of the animal obesity, a promoter of glucose intolerance[42].

A screen of major factors responsible for intestinal lipid absorption identified DGAT2 as a potential linker of BMAL1 with intestinal TAG synthesis and secretion (Fig. 5). Direct regulation of DGAT2 by BMAL1 was validated by cell-based assays and confirmed by the findings from *Rev-erbα*[iKO] mice (Figs. 6–8). Two DGATs, DGAT1 and DGAT2, are expressed in the intestine[43]. DGATs function to catalyze the conversion of DAGs to TAGs, which is the final and rate-limiting step in TAG synthesis. Both DGAT1 and DGAT2 contribute to intestinal lipid absorption, however, DGAT2 appears play a more important role because *Dgat1*[−/−] mice are viable and healthy with reduced TAG stores, but *Dgat2*[−/−] mice die in a few hours due to extremely low whole-body TAG content and an impaired skin barrier[44,45]. Humans with a *DGAT2* mutation (p.Y223H) show consistently decreased serum TAG levels[46]. Intestine-specific deletion of *Bmal1* in mice results in diet-induced obesity, that phenocopies intestine-specific knockout of and whole-body knockout of *Dgat1*[4,44]. This supports that reduced intestinal TAG synthesis is the primary impetus to attenuated obesity in HFD-fed *Bmal1*[iKO] mice. The reduction in TAG secretion from the intestine contributes to decreased circulating FFA in the mice lacking intestinal *Bmal1* (Fig. 1). It is noted that intestinal fatty acid uptake also plays a role in FFA absorption[47]. We analyzed the expression of fatty acid transporters and found that FATP4 expression was down-regulated in the conditional knockout mice, while others (CD36, FATP1 and FATP2) remained unchanged (Supplementary Fig. 8a). However, intestinal FATP4 is dispensable for dietary lipid absorption[48]. Thus, we may exclude a contribution of intestinal FFA uptake to the change in circulating FFA.

Because HFD can cause changes in gene expression in the tissues[49], we examined the intestinal expression of clock genes and the genes involved in fat absorption in HFD-fed mice (Supplementary Fig. 8). We found that intestinal BMAL1 was increased in HFD-fed wild-type mice compared to mice fed a chow diet (Supplementary Fig. 8). Likewise, DGAT2 was up-regulated by a HFD (Supplementary Fig. 8), consistent with a previous report[50]. It is likely that up-regulation of DGAT2 is secondary to the change in BMAL1 expression as BMAL1 directly drives *Dgat2* transcription. By contrast, we observed no alterations in other fat absorption-related genes including *Dgat1*, *Mogat2* and *Gpat3* (Supplementary Fig. 8). Nevertheless, HFD feeding did not blunt the reduction of DGAT2 in *Bmal1*[iKO] mice (Supplementary Fig. 8). Therefore, it should be no question that reduced DGAT2 expression underlies the metabolic phenotype of HFD-fed *Bmal1*[iKO] mice.

In conclusion, we identified the core clock gene *Bmal1* in the intestine as a key determinant of lipid homeostasis and systemic fat storage in response to excess dietary fat. Intestinal *Bmal1* promotes dietary fat absorption via regulating *Dgat2* transcription and expression. Moreover, targeted inhibition of intestinal *Bmal1* protects mice from HFD-induced obesity. These findings suggest intestinal BMAL1 as a drug target for management of metabolic diseases induced by excess fat intake.

## Methods
**Materials**. [³H]-triolein (NEC317050UC) and [¹⁴C]-oleic acid (NET431001MC) were obtained from PerkinElmer Life Sciences (Waltham, MA). SR9009 and SR8278 were purchased from MCE (Monmouth Junction, NJ). Olive oil, glucose and tyloxapol were purchased from Sigma-Aldrich (St. Louis, MO). Insulin was purchased from Procell Biotech (Newport Beach, USA). Anti-BMAL1 (1:200 dilution) and anti-GAPDH (1:10000 dilution) antibodies were purchased from Abcam (Cambridge, MA). Anti-DGAT2 (1:200 dilution) and anti-MTP (1:1000 dilution) antibodies were purchased from Santa Cruz Biotechnology (Stanta Cruz, CA). Anti-CD36 (1:1000 dilution), anti-MOGAT2 (1:1000 dilution), anti-DGAT1 (1:1000 dilution), anti-GAPT3 (1:1000 dilution), anti-PLIN3 (1:1000 dilution), anti-PPARα (1:1000 dilution), anti-ApoB (1:1000 dilution) and anti-FASN (1:1000 dilution) antibodies were obtained from Proteintech Group (Chicago, IL). Anti-rabbit IgG antibody was obtained from Cell Signaling Technology (Danvers, MA).

*Dgat2* luciferase reporters (−2000/+93, −1300/+93, −700/+93, −300/+93 bp and an E-box site-mutated version), pcDNA3.1, pcDNA3.1-Bmal1, pcDNA3.1-BMAL1, siControl (control siRNA), siBmal1 (siRNA targeting mouse *Bmal1*) and siBMAL1 (siRNA targeting human *BMAL1*) were obtained from Transheep (Shanghai, China). All siRNA sequences are provided in Supplementary Table 1.

**Animals**. Male wild-type C57BL/6 mice were obtained from HFK Biotech (Beijing, China). *Bmal1*[iKO] mice on a C57BL/6 background have been established in our laboratory[21]. The mice were identified by PCR genotyping of genomic DNA from tail biopsies (Supplementary Fig. 1a). *Rev-erbα*[iKO] mice were generated by breeding mice carrying a conditional *Rev-erbα* allele (i.e., exons 2–6 floxed allele, obtained from Cyagen Biosciences, Guangzhou, China) with mice expressing *villin-Cre*. Genotyping of mice harboring the conditional *Rev-erba* allele was performed with primers F: 5′-GCTGAACAGAGAGTCCTTCCCTGC-3′ and R: 5′-AGACTTGGGAAGGA-GATGGACAGC-3′, resulting in a 298 bp fragment for conditional allele and a 231 bp fragment for wild-type allele. Genotyping of mice expressing *villin-Cre* was performed with primers *villin-Cre*-F: 5′-GTGTTTGGTTTGGTTTCCTCTGCATAAGA-3′ and *villin-Cre*-R: 5′-GCAGGCAAATTTTGGTGTACGGTCA-3′, resulting in a 567 bp product. Mice were housed in a pathogen-free facility under a 12 h light/12 h dark cycle at controlled room temperature of 22–25 °C and a relative humidity of 40–60% with free access to water and food (if not specified).

Diets were normal chow (D12450J, Research Diets, New Brunswick, NJ) and HFD (60% calories from fat, D12492, Research Diets, New Brunswick, NJ). The animal experiments were approved by the Ethics Committee of Guangzhou University of Chinese Medicine and the experimental procedures strictly followed the guidelines for Institutional Animal Care and Use.

**HFD-induced obesity**. To determine the effect of intestinal *Bmal1* on weight gain, *Bmal1*[iKO] mice and wild-type littermates were fed HFD for 10 weeks starting from the age of 6 weeks. The control groups of mice were kept on the chow diet. Body weight was recorded weekly during the 10 weeks experiments. Body lengths were measured by subtracting the length of the tail (base to tail tip) from the length of the entire animal (snout to tail tip) after anesthesia. Body fat and lean masses were assessed using a mouse MRI (Echo Medical Systems). For experiments with time-restricted (nighttime) feeding, the mice had access to chow diet or HFD for 9–10 h in the dark period from ZT13 to ZT22/23. The food access durations were readjusted weekly to ensure isocaloric consumption in all groups (plus or minus a maximum of 1 h).

**Locomotor activity analysis**. Mice were individually housed in running wheel cages (Lafayette Instrument, Lafayette, IN) that were placed in light-tight cabinets under a 12 h light/12 h dark cycle. After acclimation to the system, mice were subjected to 2 weeks of continuous recording. Locomotor activity was analyzed using the ClockLab software (Actimetrics, Wilmette, IL). The data were separated, pooled and averaged on the basis of light/dark cycle.

**Metabolic cages**. Mice were individually housed in Promethion Metabolic Cages (Sable Systems, Las Vegas, NV) for one week before data collection. Indirect calorimetric and energy balance parameters including VO₂ (oxygen consumption, in l/min), VCO₂ (carbon dioxide expiration, in l/min) and food intake were assessed for two days. Energy expenditure (EE, in kJ/min) was calculated by using the Weir's equation (16.3 ×VO₂ + 4.6 ×VCO₂). Values of EE were normalized to the body weight raised to the power 0.75.

**Oil gavage experiments**. *Bmal1*[iKO] and control mice ($n = 6$ per group) with or without 30 min tyloxapol (500 mg/kg, i.p.) pretreatment were gavaged with olive oil (10 μl/g) at ZT2 after overnight fasting. Blood samples were collected by retro-orbital bleeding before (time 0) and at 15, 30, 60, 120, and 240 min after oil gavage. After blood sampling, mice were sacrificed and proximal jejunum and liver samples were collected. TAG and FFA were measured with specific assay kits (Jiancheng Bioengineering Institute, Nanjing, China). Additionally, fasted mice were gavaged with olive oil (10 μl/g) at each of different times of the day (ZT4, ZT8, ZT16, and ZT20). 2 h post gavage, the plasma, liver, and intestine samples were collected and the levels of TAG and FFA were measured as described above.

**OGTT and ITT**. *Bmal1*[iKO] and control mice were fasted prior to OGTT (12 h fasting) and ITT (4 h fasting) tests. For OGTT, mice ($n = 6$ per group) were gavaged with glucose (1 g/kg). Serum samples were collected by retro-orbital bleeding before (time 0) and at 15, 30, 60, 120 and 240 min after glucose gavage. For ITT, mice were intraperitoneally (i.p.) injected with insulin (1 U/kg). Serum samples were collected by retro-orbital bleeding before (time 0) and at 15, 30, 60, 120 and 240 min after insulin injection. Serum glucose was measured using the assay kit (Jiancheng Bioengineering Institute, Nanjing, China).

**SR9009 and SR8278 treatment**. 6-week-old wild-type mice were fed a HFD for 4 weeks. The mice continued on HFD and we initiated oral gavage of SR9009 (100 mg/kg) or SR8278 (100 mg/kg) or vehicle at ZT2 once daily for 4 weeks. Body weight and food intake were monitored daily. Additionally, we initiated a study

with 6-week-old mice that had been kept on chow diet. The mice were switched to HFD feeding and gavaged with SR9009 (100 mg/kg) or vehicle at ZT2 once daily for 8 weeks.

**[14C]-Oleic acid uptake and [3H]-triolein absorption**. Oleic acid uptake by the peripheral tissues and intestinal triolein absorption were examined with $Bmal1^{iKO}$ and control mice[51]. Briefly, for tissue uptake experiments, mice were fasted for 4 h and i.p. injected with 200 μl of olive oil containing 2 μCi [14C]-oleic acid. Serum samples were collected by retro-orbital bleeding before (time 0) and at 15, 30, 60, 90 and 120 min after injection. After 120 min, liver, heart, gWAT, iWAT, BAT, quadriceps and soleus were collected. To assess TAG absorption, mice were gavaged with 200 μl of olive oil containing 5 μCi [3H]-triolein after 12 h fasting during the light phase. Serum samples were collected by retro-orbital bleeding before (time 0) and at 15, 30, 60, 90 and 120 min after oil gavage. After the blood was flushed out with saline, stomach, small intestine (from the base of the stomach to the cecal junction), colon, liver, heart, gWAT, iWAT, BAT, quadriceps and soleus were collected. The small intestines were cut it into 2 cm segments. Tissues were extracted using the Folch method[52], and dissolved in a tissue solubilizer (Biosol, National Diagnostic, Atlanta, GA). Samples were then decolorized by 30% hydrogen peroxide and mixed with scintillation solution (Bioscint, National Diagnostic, Atlanta, GA). 14C and 3H contents were measured by using a liquid scintillation counter (LS 6500, Beckman Coulter, Brea, CA).

**Biochemical analysis**. TAG and FFA levels in serum and tissue samples were measured using the biochemical kits (Jiancheng Bioengineering Institute, Nanjing, China). Fecal lipids were extracted using the Folch method[52]. Briefly, samples were homogenized in chloroform: methanol (2:1, v/v), and centrifuged at 900 g for 10 min. The supernatant was collected, dried, and re-dissolved in ethanol. Lipids in the extracts were measured with the assay kits as described above. Leptin and insulin were measured by using ELISA kits (Meimian Biotechnology, Jiangsu, China). Serum chylomicron (CM) concentrations were measured by using an ELISA kit against apoB48 (Meimian Biotechnology, Jiangsu, China).

**H&E staining**. Tissues were formalin-fixed and embedded in paraffin. 5 μm paraffin-embedded sections were stained with haematoxylin & eosin (H&E). Images were captured using a AXIO Imager M1 microscope (Carl Zeiss, Oberkochen, Germany). Adipocyte size was quantified using the ImageJ software based on six sections per mouse and six mice per group.

**Oil red O staining**. Liver and proximal jejunum were fixed in 4% paraformaldehyde and embedded in Tissue-Tek O.C.T. compound. 10 μm thick sections were prepared and stained with oil red O, followed by counterstaining with hematoxylin. The red lipid droplets were visualized by a Nikon microscope (Nikon, Melville, NY).

**Immunofluorescent staining**. Intestine samples were fixed in 4% paraformaldehyde, and then transferred to a grade series of sucrose solution (10%, 20%, and 30%). The sections (20 μm thickness) were blocked with 5% BSA and 0.5% Triton X-100 in phosphate-buffered saline (PBS), and then incubated with antibodies against DGAT2 and villin. After washing with PBS, sections were sequentially incubated with secondary antibodies and with DAPI (4′,6-diamidino-2-phenylindole). Sections were washed, mounted, and imaged using a laser scanning microscope (Carl Zeiss, Oberkochen, Germany).

**Transmission electron microscopy**. Transmission electron microscopy (TEM) was used to evaluate lipid droplets in the intestine[51,53]. In brief, proximal jejunum samples were fixed in 2.5% glutaraldehyde, post-fixed in 1% $OsO_4$, dehydrated in graded ethanol and embedded in epoxy resin. Ultrathin sections (75 nm) were cut and stained with uranyl acetate and lead citrate. Images were taken by using a TEM (Hitachi HT7700, Japan).

**Cell culture and transfection**. Caco-2, CT26, and NIH3T3 cells were obtained from the American Type Culture Collection (ATCC, Manassas, VA) and cultured in Dulbecco's modified Eagle's medium (DMEM) supplemented with 10% fetal bovine serum (FBS) at 37 °C in a 5% $CO_2$ humidified atmosphere. Cells were transfected with overexpression plasmid or siRNA (Supplementary Table 1) using JetPRIME (Polyplus Transfection, Ill kirch, France). After 48 h, cells were collected for further analysis.

**Cell synchronization**. Cell synchronization was performed using a serum shock method[54]. In brief, CT26 and Caco-2 cells were seeded into a 12-well plate and transfected with siBmal1 or siBMAL1 or siControl. 12 h later, the culture medium was changed to serum-free medium. After another 12 h, 50% FBS was added for 2 h. Then, the medium was changed back to serum-free medium and cells were harvested at specific time points (24, 28, 32, 36, 40, 44, and 48 h).

**qPCR**. Total RNA was extracted using TRIzol reagent (Takara, Shiga, Japan) and reversely transcribed to cDNA using RT Master Mix (Vazyme, Nanjing, China). qPCR was performed using SYBR Green Master Mix (Vazyme, Nanjing, China) according to the manufacturer's instructions. Data were normalized to the housekeeping gene (mouse *Hmbs* or human *GAPDH*). Primers are listed in Supplementary Table 1.

**Western blotting**. Protein samples were subjected to sodium dodecyl sulfate-polyacrylamide gel electrophoresis, and then transferred to polyvinylidene fluoride membranes (Millipore, Bedford, MA). Blots were reacted with the specific primary antibodies, followed by crosslinking with the secondary antibodies. Protein bands were visualized with enhanced chemiluminescence and band densities were analyzed with FluorChem 5500 and AlphaEaseFC Softwares (Alpha Innotech, San Leandro, CA).

**Luciferase reporter assay**. NIH3T3 cells were co-transfected with *Dgat2* luciferase reporter (100 ng), pRL-nTK (a renilla luciferase reporter, 10 ng) and overexpression plasmid using JetPrime (Polyplus Transfection, Ill kirch, France). 24 h later, cells were lysed in passive lysis buffer and luciferase activities were measured using Dual-Luciferase Reporter Assay System (Promega, Madison, WI). Firefly luciferase activity was normalized to renilla luciferase activity, and expressed as relative luciferase unit (RLU).

**EMSA**. EMSA assays were performed with a chemiluminescent EMSA kit (Beyotime, Shanghai, China)[55]. Nuclear protein was incubated with biotin-labeled probe (unlabeled probe or unlabeled mutated probe was added for competitive experiments) in EMSA binding buffer. The mixture was subjected to 4% nondenaturing polyacrylamide gel electrophoresis and transferred onto a Hybond-N + membrane (Amersham, Buckinghamshire, UK). Following cross-linking, blocking, washing, and balancing, the membrane was incubated with enhanced chemiluminescent and visualized by using Omega Lum G imaging system (Aplegen, Pleasanton, CA). Oligonucleotide sequences are provided in Supplementary Table 1.

**ChIP**. ChIP assays were performed using a SimpleChip plus Enzymatic Chromatin IP kit (Cell Signaling Technology, Beverly, MA)[21]. Intestine tissues were fixed in 1% formaldehyde and lysed with lysis buffer, followed by digestion with micrococcal nuclease. Sheared chromatin was immunoprecipitated with anti-BMAL1 antibody or normal IgG (a negative control) for overnight at 4 °C. The immune complex was decross-linked at 65 °C for 4 h. The obtained DNAs were purified and analyzed by qPCR with specific primers (Supplementary Table 1).

**Statistical analyses**. Data are recorded as mean ± standard deviation (SD). The means of two groups were compared using Student's t-test. Analysis of variance (ANOVA) was performed to compare means of more than two groups. In most cases, we performed two-way repeated measures ANOVA to analyze whether the effect of genotype was significant or not. If so, post-hoc Bonferroni test was used for further group comparisons (GraphPad Prism, San Diego, CA). The level of significance was set at $p < 0.05$ (*).

## Data availability

All data generated in this study are provided in the Supplementary Information/Source Data file. ChIP-seq data were obtained from GEO database with an accession number of GSE130505. Source data are provided with this paper.

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

## Acknowledgements

This work was supported by the National Natural Science Foundation of China (81802749, 81903698), Natural Science Foundation of Guangdong Province (2021A1515011291), Guangzhou Science and Technology Project (201904010472), and the Science and Technology Foundation of Shenzhen (JCYJ20180301170047864, JCYJ20200109144410181).

## Author contributions

F.Y. and B.W. conceived and designed the study. F.Y., Z.W., T.Z., X.C., H.X., F.W., L.G., and M.C. conducted experiments. F.Y., Z.W., and K.L. performed data analysis. F.Y., K.L., and B.W. wrote and revised the manuscript.

## Competing interests

The authors declare no competing interests.
