## [Peer Review File · Nature Communications]

First round comments -

Reviewer #1 (Remarks to the Author):

The authors investigated the functional relevance of the intestine clock in dietary fat absorption and energy homeostasis and generated and tested mouse lines with intestine-specific deletion of the core clock gene *Bmal1* or *Rev-erba*. They show that the intestine clock acts as an accelerator in dietary fat absorption in response to diet rich in fat. They also propose that targeting intestinal BMAL1 may be a promising approach for management of metabolic diseases induced by excess fat intake.

The experiments were designed properly and the results are convincing.

I have no comments.

Reviewer #2 (Remarks to the Author):

The circadian rhythm has a strong influence on both body physiology and disease in humans. The master circadian clock functions from autoregulatory transcriptional, translational, and posttranslational feedback loops of few transcription factors encoded by "clock" genes, including brain and muscle aryl hydrocarbon receptor nuclear translocator-like 1 (BMAL1). The interaction of BMAL1 with CLOCK, results in the formation of a heterodimer, which binds to clock-controlled genes (CCGs) (i.e. PERs and CRYs) containing E-box element and activates their transcription. In turn, PERs and CRY protein accumulate and inhibit CLOCK: BMAL1. As the involvement of the intestine in fat transport and obesity is poorly defined, the authors designed studies using intestine-specific deletion of *Bmal1* or *Rev-erba* to investigate fat absorption and energy homeostasis. Their findings document the stimulatory role of the intestine clock in these processes.

The manuscript is well written. Its findings are interesting, innovative and challenging. The issue is fully relevant and certainly in a very critical area.

The authors have taken care to provide the rationale and justification for this extremely laborious work. However, they should address concerns and provide information to further improve the manuscript.

ABSTRACT

L-28:hyperlipidemia, glucose intolerance...are attributed to impaired intestinal lipid secretion...." Is there a strong evidence for glucose intolerance?

L-30: "Impaired intestinal lipid secretion and reduced fat absorption": The absorption of lipids is considered to precede the subsequent event (e.g. secretion of triglycerides by the chylomicron vehicle).

RESULTS

1. "Targeted deletion of *Bmal1* led to marked disruptions of rhythmic expression of other clock components (including *Rev-erba*, *Rev-erbβ*, *Per3*, *Npas2*, *Cry1*, *Cry2*, *Dbp* and *E4bp4* in the intestine". To show that this observation is specific to the intestine, did the authors analyze the expression of these genes in other organs (e.g. liver)? a matter of caution?

2. Gonadal and inguinal white adipose tissues were reported. What about the weight of mesenteric adipose tissue, which is commonly associated with cardiometabolic disorders?

3. Bmal1iKO mice exhibited decreased hepatic lipids.

Is it the consequence of a reduction in lipogenesis in the liver or of the lesser mobilization of fatty acids by adipose tissue?

4. L.150: "The mice had lower levels of intestinal lipids".

Is it on an empty stomach or after eating? Were the intestinal lipid measurements carried out in the jejunum, the optimal part for fat absorption?

5. Figure 2C (oil red staining) is not of good quality.

6. L.156: Why the authors did not measure chylomicrons, which are the best test to estimate lipid absorption and secretion after oral gavage with olive oil?

7. Figure 5D: Units are absent in the x-axis.

8. Figures 5A & 5B. Components responsible for dietary fat absorption have been assessed in small intestines to determine the mechanisms of lipid malabsorption in of Bmal1iKO mice. However, there was no estimation of intestinal fatty acid binding protein (IFABP), which is thought to be essential for the efficient uptake and trafficking of dietary fatty acids. In their study, Storch's team has reported that high-fat-fed IFABP^{-/-} mice have an increased fecal output and are likely malabsorbing other nutrients in addition to lipid (Am J Physiol Gastrointest Liver Physiol. 2020 Mar 1;318(3):G518-G530). Furthermore, they observed that the ablation of IFABP leads to marked alterations in intestinal morphology and secretory cell abundance.

9. What is the reason for choosing the mouse CT26 for the regulatory effects of BMAL1 on DGAT2? All the experiments with Bmal1 were performed with C57BL/6.

10. FoxO1 interacts with SREBP1 and influences lipid synthesis and transport, including DGAT2 expression. The authors should have taken this into consideration to further verify the mechanisms of action elicited by Bmal1 silencing.

11. In general, BMAL1 works by complexing with CLOCK. This is the heterodimer comprising BMAL1 and CLOCK, which activates the transcription of the genes encoding the PER and CRY proteins. Did the authors test the impact of BMAL1 silencing on CLOCK expression? What would we have observed in fat absorption if CLOCK had been knocked out?

DISCUSSION

L.330: "We did not analyze mice with food restricted to the light period (rest phase for mice) because intake of a regular amount of food is impossible". Don't mice only eat at night?

L.332: "Contrasting with a prior report that mice absorb lipids more efficiently in the nighttime (activity phase) based on short-term (≤ 1 h) in situ and in vitro experiments". Precisely, this is expected because of the food intake by mice especially at night.

Reviewer #3 (Remarks to the Author):

In this manuscript, the authors investigated the role of intestinal clock gene in the regulation of energy homeostasis. Using LoxP-Cre technology, mouse line with intestine-specific deletion of the clock gene Bmal1 was generated. These mice were protected against obesity and related metabolic complications after fed high-fat diet (HFD) for 10 weeks. The molecular studies revealed that the underlying mechanism was through transactivating the expression of Dgat2 gene, which encodes diglyceride acyltransferase (DGAT2) to catalyze the formation of triglycerides, essential for intestinal TAG absorption. These findings were further confirmed in a new mouse line with genetical intestinal deletion of Rev-erba gene, a known negative regulator of BMAL1. These mice had an enhanced HFD-induced obesity and comorbidities after chronic HFD feeding. In addition, the treatment with a small-molecule, SR9009, a known RVE-ERBa agonist, results in a reduced

body weight gain in mice.

The key findings are: 1) intestine clock gene acts as an accelerator in dietary fat absorption in response to diets rich in fat, and 2) DGAT2 plays an important role in mediating clock gene-induced triglycerides uptake. These findings are innovative and are interesting in the fields of nutrition and metabolism. In this manuscript, most of materials and experimental procedures were described in sufficient detail. Most of the conclusions are supported by their observations. While the findings obtained from the experiments are interesting, some substantial concerns are identified.

Major concerns:

1. The investigators used 3H-triolein to determine how the clock gene affect the triglyceride absorption from the intestine. However, there are several potential problems:

a. It is unclear if the investigators flushed out the blood from the body before the tissue samples were collected and analyzed. The residual blood within the tissues could significantly contribute and therefore affect the tissue radioactive recovery. (The same concern also existed about the experiment with 14C-Oleic acid because it is unknown if the blood was removed before the tissue samples were collected).

b. When the authors collected tissue samples, they missed two important tissues for measuring the counts (cpm) of 3H-triolein. One is the stomach because the 3H-triolein was gavaged into stomach. The reduced counts in serum or other tissues might be simply due to slower stomach empty in the *Bmal1*^{-/-} mice. The 2nd tissue is the colon. The *Bmal1*^{-/-} mice might have a more rapid intestinal motility, so the gavaged 3H-triolein could have quickly passed the small intestine and entered the colon, where did not make chylomicron, resulting in a reduction of the 3H-triolein transport.

c. The faeces should also be collected during the 3H-triolein experiment and used for measuring the radioactivity. Without that data and without the colon counts, the authors were unable to find out where the administered 3H-triolein was.

2. Based on the description in the manuscript, "the tissues were minced and dissolved in a tissue solubilizer (Biosol), decolorized by 30% hydrogen peroxide and mixed with scintillation solution", one unanswered question is how confident the authors are that the measurement (cpm) was from the intact 3H-triolein because the radioactivity (cpm) could include both lipid soluble as well as water soluble moieties, e.g., detached 3H and metabolites of 3H-triolein. A proper way is to use Folch method to exact the lipid soluble radioactivity before liquid scintillation counting.

3. It does not make sense to use such high doses of 14C-oleic acid (2 mCi) and 3H-triolein (5mCi) in each mouse. This must be an error as this is a lot of radioactivity.

4. The data presented in this manuscript showed that DGAT2 plays a very important role in mediating clock gene-induced triglycerides uptake. There are two isozymes of DGAT encoded by the genes *DGAT1* and *DGAT2*. It has been known that the *DGAT1* is mainly located in absorptive enterocyte cells that line the intestine and duodenum. However, *DGAT2* is mainly located in fat, liver and skin cells. Even in the published paper cited by the authors has clearly described that "Expression levels for human small intestine were very low". If this is the case, the question is how relevant if the reported data with regard to humans.

5. Rodents normally eat the food during the dark phase (nighttime). It would be more interesting to test whether the time-restricted feeding during the day time (light phase), instead of nighttime, affects HFD-induced obesity in a *Bmal1*-depende manner or not. While the mice with light-time food restriction may not consume the same amount of HFD as the mice without any food restriction, light-time restriction will allow to determine if the shift of feeding time alters intestinal clock gene expression and related metabolism.

6. *DGAT1* inhibitors have been proposed to treat obesity and a number of *DGAT-1* inhibitors were in clinical trials for this indication. However, recent findings raised the concern for *DGAT1* inhibition in humans because of the severe side effects which include nausea, diarrhea, and vomiting following meals containing fat (Hass, et al. *J Clin Invest.* 2012). Did the authors noticed if the *Dgat2* knockout mice had any diarrhea during the experiment?

Minor concerns:

1). The method how to measure the fecal fat should be described.

2). The description in "Statistical analyses" is not detail enough because many of data in each of the figures should be analyzed with Two-Way Repeated Measures ANOVA.

3). More information regarding animal model should be provided, e.g. the source of *Rev-erbaloxp* mice.

We wish to thank the reviewers for careful and valuable reviews. Below are our point-by-point responses to the reviewer comments.

Responses to reviewer #1

The authors investigated the functional relevance of the intestine clock in dietary fat absorption and energy homeostasis and generated and tested mouse lines with intestine-specific deletion of the core clock gene *Bmal1* or *Rev-erba*. They show that the intestine clock acts as an accelerator in dietary fat absorption in response to diet rich in fat. They also propose that targeting intestinal BMAL1 may be a promising approach for management of metabolic diseases induced by excess fat intake.

The experiments were designed properly, and the results are convincing.

I have no comments.

Response: N/A

Responses to reviewer #2

The circadian rhythm has a strong influence on both body physiology and disease in humans. The master circadian clock functions from autoregulatory transcriptional, translational, and posttranslational feedback loops of few transcription factors encoded by “clock” genes, including brain and muscle aryl hydrocarbon receptor nuclear translocator-like 1 (BMAL1). The interaction of BMAL1 with CLOCK, results in the formation of a heterodimer, which binds to clock-controlled genes (CCGs) (ie. PERs and CRYs) containing E-box element and activates their transcription. In turn, PERs and CRYs protein accumulate and inhibit CLOCK: BMAL1. As the involvement of the intestine in fat transport and obesity is poorly defined, the authors designed

studies using intestine-specific deletion of Bmal1 or Rev-erba to investigate fat absorption and energy homeostasis. Their findings document the stimulatory role of the intestine clock in these processes.

The manuscript is well written. Its findings are interesting, innovative and challenging. The issue is fully relevant and certainly in a very critical area.

Response: Thanks very much for the nice comments on our work.

The authors have taken care to provide the rationale and justification for this extremely laborious work. However, they should address concerns and provide information to further improve the manuscript.

ABSTRACT

L-28: "...hyperlipidemia, glucose intolerance...are attributed to impaired intestinal lipid secretion...." Is there a strong evidence for glucose intolerance?

Response: We understand the reviewer's concern. Glucose intolerance, associated with obesity, is not a direct effect of impaired intestinal lipid secretion. Accordingly, "glucose intolerance" has been removed from the sentence in the revised version.

L-30: "Impaired intestinal lipid secretion and reduced fat absorption": The absorption of lipids is considered to precede the subsequent event (eg. secretion of triglycerides by the chylomicron vehicle).

Response: We accept the reviewer's criticism. Accordingly, we have changed the statement to "impaired lipid resynthesis in the intestine and reduced fat secretion" in the revised manuscript.

RESULTS

1. “Targeted deletion of *Bmal1* led to marked disruptions of rhythmic expression of other clock components (including *Rev-erba*, *Rev-erbβ*, *Per3*, *Npas2*, *Cry1*, *Cry2*, *Dbp* and *E4bp4* in the intestine”. To show that this observation is specific to the intestine, did the authors analyze the expression of these genes in other organs (e.g. liver)? a matter of caution?

Response: The reviewer raised a comment that the expression of clock genes should be analyzed in other organs such as the liver in *Bmal1^{iKO}* mice. In fact, we have already shown that intestine-specific knockout of *Bmal1* has no effects on *Bmal1* expression in other organs including the liver, kidney, brain and adipose tissues (data are shown in Fig. S1b). To alleviate the reviewer’s concern, we have performed new experiments to analyze the expression of clock genes (including *Rev-erba*, *Rev-erbβ*, *Per3*, *Npas2*, *Cry1*, *Cry2*, *Dbp* and *E4bp4*) in the livers of both *Bmal1^{iKO}* and control mice, and found no changes in hepatic expression of these genes. The new data have been added to Fig. S2 in the revised manuscript.

2. Gonadal and inguinal white adipose tissues were reported. What about the weight of mesenteric adipose tissue, which is commonly associated with cardiometabolic disorders?

Response: We agree with the reviewer that it is insightful to also test mesenteric adipose tissue. Accordingly, we have performed new experiments to analyze the weight changes in mesenteric adipose tissues (mWAT). The results showed that mesenteric adipose tissue was lighter in *Bmal1^{iKO}* mice than in control mice fed on HFD. The new data have been added to Fig 1e.

3. *Bmal1*^{iKO} mice exhibited decreased hepatic lipids. Is it the consequence of a reduction in lipogenesis in the liver or of the lesser mobilization of fatty acids by adipose tissue?

Response: The reviewer raised a concern whether lipogenesis and fatty acid mobilization play a role in decreased hepatic lipids in *Bmal1*^{iKO} mice. We have performed new experiments to assay the expression of lipogenesis-related genes and fatty acid mobilization-related genes of two genotypes. We found that the expression of these genes was unaffected in *Bmal1*^{iKO} mice. Thus, we may exclude a role of lipogenesis and fatty acid mobilization in decreasing hepatic lipids. The new data have been added to Fig. S6, and relevant result description to the main text.

4. L.150: "The mice had lower levels of intestinal lipids".

Is it on an empty stomach or after eating? Were the intestinal lipid measurements carried out in the jejunum, the optimal part for fat absorption?

Response: Intestinal lipids were measured on an empty stomach as the mice were fasted overnight (12 h) prior to sampling. The reviewer is right that the intestinal lipid measurements were carried out in the jejunum.

5. Figure 2C (oil red staining) is not of good quality.

Response: We have improved the quality of Fig 2c (please see the new Fig 2c).

6. L.156: Why the authors did not measure chylomicrons, which are the best test to

estimate lipid absorption and secretion after oral gavage with olive oil?

Response: Good suggestion. We have performed ELISA assays to measure serum chylomicrons. The new data have been added to Fig 2i in the revised manuscript.

7. Figure 5D: Units are absent in the x-axis.

Response: Fixed.

8. Figures 5A & 5B. Components responsible for dietary fat absorption have been assessed in small intestines to determine the mechanisms of lipid malabsorption in of *Bmal1*^{IKO} mice. However, there was no estimation of intestinal fatty acid binding protein (IFABP), which is thought to be essential for the efficient uptake and trafficking of dietary fatty acids. In their study, Storch's team has reported that high-fat-fed IFABP^{-/-} mice have an increased fecal output and are likely malabsorbing other nutrients in addition to lipid (Am J Physiol Gastrointest Liver Physiol. 2020 Mar 1;318(3): G518-G530). Furthermore, they observed that the ablation of IFABP leads to marked alterations in intestinal morphology and secretory cell abundance.

Response: The reviewer raised a comment that the authors need to estimate intestinal fatty acid binding protein (IFABP). In fact, we have already tested it and data are shown in Fig 5a (*Fabp2*). FABP2 is also known as IFABP. There is no significant difference in intestinal *Fabp2* (IFABP) expression between *Bmal1*^{IKO} and control mice (Fig 5a). Thus, involvement of *Fabp2* (IFABP) in lipid malabsorption in *Bmal1*^{IKO} mice may be excluded.

9. What is the reason for choosing the mouse CT26 for the regulatory effects of

BMAL1 on DGAT2? All the experiments with *Bmal1* were performed with C57BL/6.

Response: We used CT26 cell model to study the regulatory effects of BMAL1 on DGAT2 because CT26 is a mouse colon carcinoma cell line that can be passaged stably (relatively easy to cultivate) and have stable traits. To alleviate the reviewer's concern, we have performed the regulatory experiments using normal cells, namely, mouse primary intestinal epithelial cells. We observed similar regulatory effects of BMAL1 on DGAT2 in normal cells. The new data have been added to Fig. S10.

10. FoxO1 interacts with SREBP1 and influences lipid synthesis and transport, including DGAT2 expression. The authors should have taken this into consideration to further verify the mechanisms of action elicited by *Bmal1* silencing.

Response: Good suggestion. We have performed new experiments and examined the expression of *Foxo1* and *Srebp1* in *Bmal1*^{ikO} versus control mice. The KO mice did not show changes in *Foxo1* or *Srebp1* expression. The new data have been added to Fig. S7 in the revised manuscript.

11. In general, BMAL1 works by complexing with CLOCK. This is the heterodimer comprising BMAL1 and CLOCK, which activates the transcription of the genes encoding the PER and CRY proteins. Did the authors test the impact of BMAL1 silencing on CLOCK expression? What would we have observed in fat absorption if CLOCK had been knocked out?

Response: Yes. We have already analyzed *Clock* expression in small intestine of *Bmal1*^{ikO} mice (please see Fig. S1d). We have found that intestinal ablation of BMAL1 did not alter *Clock* expression (two-way ANOVA). The reviewer further raised a

question what would happen if CLOCK was knocked out. As noted by the reviewer, CLOCK protein is a partner of BMAL1 in regulating gene expression via E-box elements. We show that BMAL1 transactivates the *Dgat2* gene via direct binding to an E-box in the promoter, thereby promoting dietary fat absorption. It is conceivable that CLOCK also acts on this E-box to activate *Dgat2* expression to promote fat absorption. However, whether CLOCK truly regulates fat absorption in a similar manner to BMAL1 requires experimental validations.

DISCUSSION

L.330: “We did not analyze mice with food restricted to the light period (rest phase for mice) because intake of a regular amount of food is impossible”. Don't mice only eat at night?

Response: It may be not clear to the reviewer that mice do not eat at night only (Nat Med. 2012;18(12):1768-77 // Cell Metab. 2012;15(6):848-60 // Cell Metab. 2007;6(5):414-21). Mice show a robust diurnal rhythm in food intake with ~80% of total daily food intake in the dark period and ~20% in the light period (Cell Metab. 2007;6(5):414-21).

L.332: “Contrasting with a prior report that mice absorb lipids more efficiently in the nighttime (activity phase) based on short-term (≤ 1 h) in situ and in vitro experiments”. Precisely, this is expected because of the food intake by mice especially at night.

Response: The reviewer raised a comment that it is expected that mice absorb lipids more efficiently in the nighttime because of the food intake by mice especially at night. We respectfully disagree with the reviewer on this matter. “Efficiency” here primarily measures the rate of absorption, independent of food amount provided. Absorption efficiency depends on multiple factors including the efficiencies in lipid uptake,

re-synthesis, and secretion. Therefore, more food intake is not equivalent to more efficient (rapid) absorption. In other words, it may be not valid that food intake by mice especially at night will translate to more efficient absorption in the nighttime.

Responses to reviewer #3

In this manuscript, the authors investigated the role of intestinal clock gene in the regulation of energy homeostasis. Using LoxP-Cre technology, mouse line with intestine-specific deletion of the clock gene *Bmal1* was generated. These mice were protected against obesity and related metabolic complications after fed high-fat diet (HFD) for 10 weeks. The molecular studies revealed that the underlying mechanism was through transactivating the expression of *Dgat2* gene, which encodes diglyceride acyltransferase (DGAT2) to catalyze the formation of triglycerides, essential for intestinal TAG absorption. These findings were further confirmed in a new mouse line with genetical intestinal deletion of *Rev-erb α* gene, a known negative regulator of BMAL1. These mice had an enhanced HFD-induced obesity and comorbidities after chronic HFD feeding. In addition, the treatment with a small-molecule, SR9009, a known RVE-ERB α agonist, results in a reduced body weight gain in mice.

The key findings are: 1) intestine clock gene acts as an accelerator in dietary fat absorption in response to diets rich in fat, and 2) DGAT2 plays an important role in mediating clock gene-induced triglycerides uptake. These findings are innovative and are interesting in the fields of nutrition and metabolism. In this manuscript, most of materials and experimental procedures were described in sufficient detail. Most of the conclusions are supported by their observations. While the findings obtained from the experiments are interesting, some substantial concerns are identified.

Response: Thanks very much for nice summary of the key points of our work.

Major concerns:

1. The investigators used ^3H -triolein to determine how the clock gene affect the triglyceride absorption from the intestine. However, there are several potential problems:

a. It is unclear if the investigators flushed out the blood from the body before the tissue samples were collected and analyzed. The residual blood within the tissues could significantly contribute and therefore affect the tissue radioactive recovery. (The same concern also existed about the experiment with ^{14}C -Oleic acid because it is unknown if the blood was removed before the tissue samples were collected).

Response: We regret that the blood flushing details in [^{14}C]-oleic acid uptake and [^3H]-triolein absorption experiments were missing in our original submission. In fact, we did flush out the blood from the body using ice-cold saline before collection of tissue samples. This information has been added to the *Methods* section (under “[^{14}C]-oleic acid uptake and [^3H]-triolein absorption”) in the revised manuscript.

b. When the authors collected tissue samples, they missed two important tissues for measuring the counts (cpm) of ^3H -triolein. One is the stomach because the ^3H -triolein was gavaged into stomach. The reduced counts in serum or other tissues might be simply due to slower stomach empty in the *Bmal1*^{-/-} mice. The 2nd tissue is the colon. The *Bmal1*^{-/-} mice might have a more rapid intestinal motility, so the gavaged ^3H -triolein could have quickly passed the small intestine and entered the colon, where did not make chylomicron, resulting in a reduction of the ^3H -triolein transport.

c. The feces should also be collected during the ^3H -triolein experiment and used for measuring the radioactivity. Without that data and without the colon counts, the authors were unable to find out where the administered ^3H -triolein was.

Response: We agree with the reviewer that it is insightful to also measure the counts

of [³H]-triolein in the stomach, colon and feces. Accordingly, new experiments have been performed and [³H] counts in the stomach, colon and feces have been analyzed. We have also performed *in vivo* gut transit experiments (by gavaging 5% Evans blue, a non-absorbable colored marker) to determine and compare gut transit time in *Bmal1^{iKO}* versus control mice. [³H] counts were not different in the stomach between two genotypes. By contrast, *Bmal1^{iKO}* mice showed a higher level, but not statistically significant, of colonic [³H]. Fecal [³H] was significantly higher in *Bmal1^{iKO}* than in control mice. In addition, gut transit time (reflective of intestinal motility) was not different between two genotypes. These findings exclude a role of stomach emptying and gut motility in reduced intestinal lipid absorption in *Bmal1^{iKO}* mice. The slightly higher level of colonic [³H] may result from transit of more [³H]-oleic acid (and its derivatives) to the colon caused by reduced absorption of [³H]-triolein in small intestine. Please note that the colon is incapable of absorbing long-chain fatty acids such as oleic acid and has an ability to absorb short- and medium-chain fatty acids such as oleic acid metabolites (Gastroenterology. 2001;120(5):1152-61 // J Lipid Res. 2016;57(6):943-54 // Gut. 1998; 43:478-83 // Am J Dig Dis. 1966;11(6):474-9 // J Appl Physiol. 1966; 21:1059-62). The new data of [³H] counts in the stomach, colon and feces have been added to Fig 4e, and new data of gut transit time to Fig. S1c.

2. Based on the description in the manuscript, “the tissues were minced and dissolved in a tissue solubilizer (Biosol), decolorized by 30% hydrogen peroxide and mixed with scintillation solution”, one unanswered question is how confident the authors are that the measurement (cpm) was from the intact 3H-triolein because the radioactivity (cpm) could include both lipid soluble as well as water soluble moieties, e.g., detached 3H and metabolites of 3H-triolein. A proper way is to use Folch method to exact the lipid soluble radioactivity before liquid scintillation counting.

Response: We agree with the reviewer that Folch method is more appropriate to extract the lipid soluble radioactivity. Accordingly, we have re-performed the [¹⁴C]-oleic

acid uptake and [³H]-triolein absorption experiments, in which the samples were extracted using the Folch method (please see the revised methods, under “[¹⁴C]-oleic acid uptake and [³H]-triolein absorption”) before liquid scintillation counting. The data have been updated accordingly.

3. It does not make sense to use such high doses of ¹⁴C-oleic acid (2 mCi) and ³H-triolein (5mCi) in each mouse. This must be an error as this is a lot of radioactivity.

Response: We regret for a mistake in the unit of radioactivity. In fact, the doses of 2 and 5 μCi were used for ¹⁴C-oleic acid and ³H-triolein, respectively. This has been fixed in the revised manuscript. Thanks very much for the reviewer’s careful review.

4. The data presented in this manuscript showed that DGAT2 plays a very important role in mediating clock gene-induced triglycerides uptake. There are two isozymes of DGAT encoded by the genes DGAT1 and DGAT2. It has been known that the DGAT1 is mainly located in absorptive enterocyte cells that line the intestine and duodenum. However, DGAT2 is mainly located in fat, liver and skin cells. Even in the published paper cited by the authors has clearly described that “Expression levels for human small intestine were very low”. If this is the case, the question is how relevant if the reported data with regard to humans.

Response: The reviewer is right that the early studies by the group of Dr. Farese suggest that DGAT2 (mRNA) is expressed at a very low level in human intestine (J Biol Chem. 2001;276(42):38870-6 // J Clin Invest. 2012;122(12):4680-4). However, this expression finding requires further validations as the scientists commented in their publications that “however, a thorough evaluation of DGAT2 expression in human enterocytes has not been done” and that “a thorough evaluation of DGAT2 expression in human enterocytes has not been elucidated” (Biochim Biophys Acta.

2013;1831(8):1377-85 // Natural Product Communications. 2018; 13(4):471-474).

In a recent study, van Rijn *et al* demonstrated that DGAT2 protein is expressed in human small intestine and human intestinal organoids (J Lipid Res. 2019;60(10):1787-1800, please refer to Fig. 2 in this paper for detailed information). The authors indicate that DGAT2 could have a functional role in the human intestinal stem cell niche and may serve as a therapeutic target to treat diseases related to intestinal lipid uptake. Therefore, it should be of little concern that the role of DGAT2 in mediating clock gene-induced TAG uptake is relevant to humans.

5. Rodents normally eat the food during the dark phase (nighttime). It would be more interesting to test whether the time-restricted feeding during the day time (light phase), instead of nighttime, affects HFD-induced obesity in a Bmal1-depende manner or not. While the mice with light-time food restriction may not consume the same amount of HFD as the mice without any food restriction, light-time restriction will allow to determine if the shift of feeding time alters intestinal clock gene expression and related metabolism.

Response: The reviewer commented that rodents normally eat the food during the dark phase (nighttime). We would like to draw the reviewer's attention to the fact that mice do not eat at night only (Nat Med. 2012;18(12):1768-77 // Cell Metab. 2012;15(6):848-60 // Cell Metab. 2007;6(5):414-21). In fact, mice show a robust diurnal rhythm in food intake with ~80% of total daily food intake in the dark period and ~20% in the light period (Cell Metab. 2007;6(5):414-21).

We agree with the reviewer that it is interesting to test whether time-restricted feeding during the daytime (light phase, the rest phase for mice) affects HFD-induced obesity. As also noted by the reviewer, mice with light-time food restriction may not consume the same amount of food as the mice without any food restriction. In this case, effects

of feeding time on HFD-induced obesity cannot be accurately assessed. To circumvent this problem, in our new experiments, we have adopted an initial adjustment period (days 1–15) that allows the mice to adapt to the time-restricted access to food before introducing the mice to a same amount of HFD. We found that mice fed during the light period gained significantly more weight and were heavier compared to mice fed *ad libitum* and mice during the dark period. However, these differences were not seen in *Bmal1^{IKO}* mice, supporting a critical role of intestinal BMAL1 in circadian fat absorption and HFD-induced obesity. The new data have been added to Fig. S14 in the revised manuscript.

6. DGAT1 inhibitors have been proposed to treat obesity and numbers of DGAT-1 inhibitors were in clinical trials for this indication. However, recent findings raised the concern for DGAT1 inhibition in humans because of the severe side effects which include nausea, diarrhea, and vomiting following meals containing fat (Hass, et al. J Clin Invest. 2012). Did the authors notice if the *Dgat2* knockout mice had any diarrhea during the experiment?

Response: The reviewer is right that the major adverse effects of DGAT1 inhibitors in the clinic relate to gastrointestinal effects (nausea, diarrhea, and vomiting). However, the side effects of certain DGAT1 inhibitors can be tolerated for most people. A good example refers to icosapent ethyl [Vascepa[®] or ethyl eicosapentaenoic acid, a highly purified omega-3 fatty acid and DGAT inhibitor (Lipids Health Dis. 2016;15(1):118 // Am J Cardiovasc Drugs. 2014;14(6):471-8 // Expert Opin Inv Drug. 2016;25(12):1457-63 // Expert Rev Cardiovasc Ther. 2020;18(4):175-80 // Postgrad Med. 2014;126(7):7-18 // Am J Cardiovasc Drugs. 2020: 1-8)], a drug approved by the US Food and Drug Administration as an adjunct to diet to reduce triglyceride levels in adult patients with severe hypertriglyceridemia.

The reviewer raised a question if the *Dgat2* knockout mice had any diarrhea during

the experiment? We are unable to provide a precise answer to this question because *Dgat2*^{-/-} mice die in a few hours due to extremely low whole-body TAG content and an impaired skin barrier (J Biol Chem. 2004;279(12):11767-76). However, we may hold the opinion that moderate rather than complete inhibition of intestinal DGAT activity is therapeutically beneficial for metabolic diseases induced by excess fat intake, while minimizing potential side effects.

Minor concerns:

1). The method how to measure the fecal fat should be described.

Response: Revised as suggested.

2). The description in “Statistical analyses” is not detail enough because many of data in each of the figures should be analyzed with Two-Way Repeated Measures ANOVA.

Response: Revised as suggested.

3). More information regarding animal model should be provided, eg. the source of Rev-erbaloxp mice.

Response: Revised as suggested.

Reviewer comments, second round –

Reviewer #2 (Remarks to the Author):

No further concerns

Reviewer #3 (Remarks to the Author):

The authors have adequately addressed my comments with new experimental data and revised the manuscript accordingly. As a result, the revised manuscript has been improved considerably over the previous submission. However, there is a concern and an error, which should be addressed.

Concern:

1. In Fig. 2i, it is unclear how the serum chylomicrons (CM) was measured. This is related to the response to Reivewer-2's comment #6. More information should be provided regarding the ELISA assay to measure serum CM.

An error:

2. In should be (Fig. 5a and Fig. S8) in Page 10, line 207

Responses to reviewer #3

The authors have adequately addressed my comments with new experimental data and revised the manuscript accordingly. As a result, the revised manuscript has been improved considerably over the previous submission. However, there is a concern and an error, which should be addressed.

Concern:

1. In Fig. 2i, it is unclear how the serum chylomicrons (CM) were measured. This is related to the response to Reviewer-2's comment #6. More information should be provided regarding the ELISA assay to measure serum CM.

Response: Revised as suggested (please see the Methods under "*Biochemical analysis*").

An error:

2. In should be (Fig. 5a and Fig. S8) in Page 10, line 207

Response: Fixed.